# Failure of Marmolada Glacier (Dolomites, Italy) in 2022: Data-based back analysis of possible collapse mechanisms

Roberto G. Francese[1,2], Roberto Valentino[1], Wilfried Haeberli[3], Aldino Bondesan[4,5], Massimo Giorgi[2], Stefano Picotti[2], Franco Pettenati[2], Denis Sandron[2], Gianni Ramponi[6], and Mauro Valt[7]

[1]Department of Chemistry, Life Sciences and Environmental Sustainability, University of Parma, Parma, 43124, Italy
[2]National Institute of Oceanography and Applied Geophysics – OGS, Sgonico, 34010, Italy
[3]Department of Geography, University of Zurich, Zurich, 8057, Switzerland
[4]Department of Historical and Geographic Sciences and the Ancient World, University of Padova, Padova, 35122, Italy
[5]Department of Military Geography, University of Stellenbosch, Stellenbosch, 7602, South Africa
[6]Department of Engineering and Architecture, University of Trieste, Trieste, 34127, Italy
[7]Environmental Protection Agency of Veneto, Centro Valanghe di Arabba, Arabba, 32020, Italy

*Correspondence to*: Aldino Bondesan (aldino.bondesan@unipd.it)

## Abstract

A small, isolated portion of the Marmolada glacier broke off on July 3, 2022. The detached ice mass had an estimated volume of 70,400 $m^3$ and slid down the slope killing 11 mountaineers after having travelled for approximately 2.3 km along the northern slope. This event is considered among the deadliest ice avalanches historically recorded in the Alps.

The unusually high air temperatures in late spring and early summer of that year led to an excess of meltwater, which, since mid-June, overpressurized the englacial discharge network, partly blocked due to frozen conditions at its base. Ice temperature, subglacial permafrost, and heat exchange from meltwater were among the primary factors controlling the thermal state of the sliding surface.

The cause of the collapse was investigated by exploiting a conceptual model that was further corroborated through simplified numerical simulations using the Limit Equilibrium Method. Pre- and post-failure satellite and aerial images, laser mapping, geophysics and morpho-climatic data were gathered in a comprehensive database and analysed to better understand the role and interaction of the predisposing and triggering factors as well as their mutual interaction. Particular attention was given to reconstructing the varying conditions of the failure surface, which partly developed along ice foliations near the glacieret's base and partly right at the ice-bedrock interface. An earthquake triggering the failure was excluded based on the processing of the available seismological observations.

It resulted that none of the three forces considered in the numerical analysis—namely, hydrostatic pressure in crevasses, hydraulic jacking, and basal friction reduction—individually caused the condition of instability. To reach this condition, it was necessary to invoke a combination of these actions, for which it was finally possible to estimate their relative weights.

## 1 Introduction

The catastrophic collapse of the Marmolada Glacier (Dolomites, Italy) fits into the broader worldwide context of mountain glaciers that are experiencing unprecedented rates of retreat associated with climate change (Hock et al., 2019; Marzeion et al., 2020; Pelto, 2020; Marta et al., 2021; Rounce et al., 2023; Zemp et al., 2019). Alpine glaciers retreat has markedly accelerated since the late 1980s (Haeberli and Hoelzle, 1995; Paul et al., 2004; Zemp et al., 2015), with collapse events increasing in frequency since the early 2000s (Chiarle et al., 2022). This acceleration has further been enhanced by rapid feedback effects, such as the degradation of permafrost (permanently-frozen ground) occurring at a global scale (e.g., Biskaborn et al., 2019; Rossi et al., 2022), which plays an increasingly important role on the stability of glaciers (e.g., Allen et al., 2022; Noetzli and Gruber, 2009).

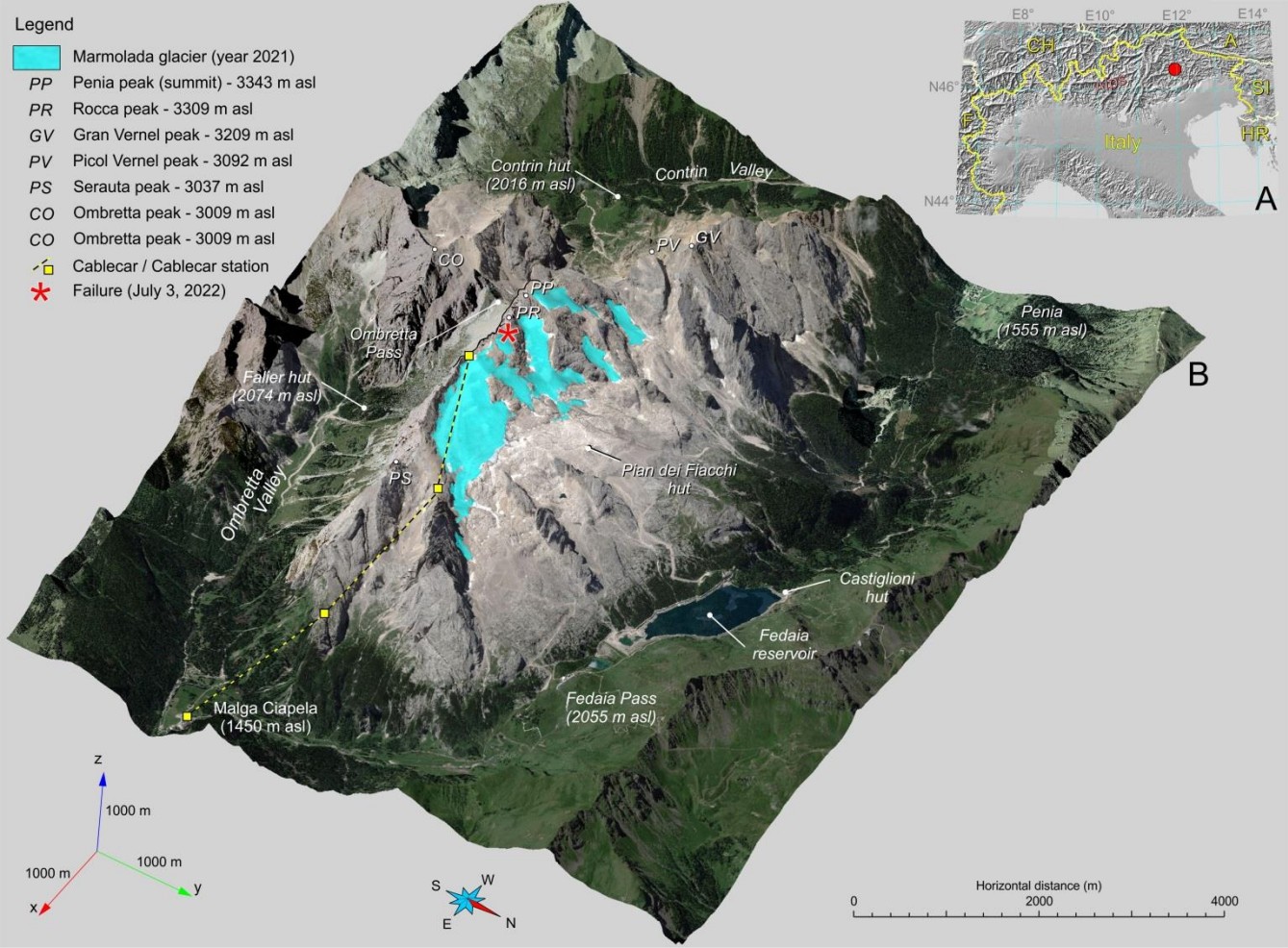

**Figure 1: (A) Alpine chain with marked position of the Marmolada glacier in the north-eastern Alps in Italy. (B) Orthomosaic of satellite and aerial images of the Marmolada massif draped on a 5m by 5m digital terrain model. The lettering mark topographic and anthropic features. The failure site is marked with a red asterisk. Aerial and satellite imagery source: Pléiades Neo, AIRBUS Defence & Space; Autonomous Province of Trento (PAT); ESRI world imagery.**

The magnitude of glacial collapse events (Alean, 1985; Dutto et al., 1991; Margreth et al., 2011; Gilbert et al., 2015; Lützow et al., 2023) could be really harmful (Tian et al., 2017). The massive failure at Juuku pass in Kyrgyzstan, on July 9, 2022, involved an ice volume three times larger than the Marmolada case (Gascoin and Berthier, 2023) and fortunately occurred in an uninhabited landscape. Similar events are documented in the French and Swiss Alps (Vincent et al., 2015; Bodin et al., 2017; Egli et al., 2021), in the Central Andes of Argentina (Falaschi et al., 2019) or in the Tibetan Plateau (Zhao et al., 2022). The significant glacier retreat and the formation of new glacial lakes is posing further potential threats such as outbursts and floods (Aggarwal et al., 2016; Haeberli et al., 2017; Huggel et al., 2002; Viani et al., 2020).

Rock, rock-ice and ice avalanches from steep and cold mountain slopes often start from detachment zones affected by permafrost and cold or polythermal ice partially or totally frozen to its bed. These aspects can involve complex, climate-related thermo-hydraulic interactions (Haeberli et al., 2004a; Margreth et al., 2017), which must be considered in integrative hazard analyses and assessments (Allen et al., 2022). The large amount of energy instantly released during such failures (Noetzli et al., 2006; Bessette-Kirton and Coe, 2020), often with little to no evident warning signs, emphasizes the importance (Margreth et al., 2011; 2017) of identifying and understanding the key factors driving instability.

Despite the growing body of research on glacial collapses (Haeberli et al., 2004b; Pralong and Funk, 2006; Faillettaz et al., 2011), significant gaps remain in our knowledge regarding the interplay of various factors contributing to instability. In addition the variables controlling ice avalanche rheology are poorly known (Hutter, 1997; Faillettaz et al., 2015; Thibert et al., 2018).

This study aims to partially address this gap with a dual objective: first, to identify the controlling factors of the collapse and classify them as predisposing or triggering; and second, to develop a numerical model to assess the relative influence of the different triggering factors. Multiple variables contributed to the catastrophic collapse that claimed several lives on July 3, 2022 (Chiarle et al., 2022; Olivieri and Bettanini, 2022; Bondesan and Francese, 2023). This required the collection and reorganization of various data into a comprehensive digital database to gain deeper insight.

There are few papers dealing with numerical simulation of glacial collapses (Jouvet et al., 2011; Logan et al., 2017; Löfgren et al., 2023) mostly due to the difficulties of inferring the appropriate physical and mechanical parameters of the ice and of the ice-bedrock contact and secondly because of the relatively rare occurrence of such events. The complex relationship between frictional sliding, water infiltration and tensional cracking (Faillettaz et al., 2011; Stearns and Van Der Veen, 2018) affects the effectiveness of deterministic approaches focused on forecasting hazards (Faillettaz et al., 2015).

We used a data-based back-analysis approach to infer the basal properties of the failure surface, in order to understand the critical interactions among englacial water (which altered temperature and pressure fields within the glacier and at its base), permafrost in rocks and sediments (Noetzli and Gruber, 2009; Rossi et al., 2022), the glacier's thermal state, and the possible existence of a thin, heterogeneous, and discontinuous layer at the ice-bedrock interface (Zoet and Iverson, 2020; Huang et al., 2024).

Numerical simulations were conducted by means of the Limit Equilibrium Method (LEM), which is routinely used for slope stability analyses (Saim and Kasa, 2023) in geotechnical engineering. Particular attention was given in defining the properties characterizing the interactions between the materials surrounding the ice-rock interface, as well as the geometry and physical properties of the ice body. The purpose was achieved by re-processing and carefully analysing both pre- and post-failure RES (Radio-Echo Sounding) profiles (Fretwell et al., 2013; Francese et al., 2019), which contributed to the conceptualization of the model for numerical simulations. Pre- and post-failure aerial and satellite imagery as well as aerial and terrestrial laser data further contributed to this development. Available meteorological data (air temperature, rainfall and snow cover) as well as cryospheric data (permafrost and ice temperature) were carefully analysed. Finally, seismological observations were included to assess the possibility of an earthquake triggering the failure.

The overall structure of the paper is as follows: section 2 provides information on geology, morphology and glaciology of the Marmolada massif; section 3 describes the various data types considered for the study along with the different processing methods; section 4 examines the temporal and spatial evolution of the factors responsible for the collapse and introduces the outcome of the numerical simulations used to evaluate the glacier stability; finally, an ample discussion is provided in section 5 on the importance of the various factors; their complex mutual relationships are modelled in the numerical simulations.

## 2 General Settings

The Marmolada Glacier (ID 941, Italian Glacier Inventory) represents a prominent location within the Dolomites in northeastern Italy (46°26′32″ N, 11°51′53″ E). This glacier drapes the northern flanks of the Marmolada massif (Fig. 1), the Dolomites' highest peak, reaching an elevation of 3343 m above sea level (asl).

### 2.1 Geomorphological and geological framework

The Marmolada massif features a multifaceted geological structure dominated by Ladinian limestones (Calcare della Marmolada), which forms part of the Dolomia dello Sciliar formation (Antonelli et al., 1990; Bosellini, 1996). This formation encapsulates a rich record of the region's geological evolution, ranging from Palaeozoic volcanic activity to the development of Triassic carbonate platforms. Structurally, the massif exhibits a monoclinal arrangement, with bedding planes dipping northward, creating a stepped "cuesta" morphology. The massif is bordered by several significant valleys, including the Ombretta, Contrin, Avisio, and Pettorina valleys, which define its margins and modulate local glacial dynamics. A striking feature along the southern side is a nearly vertical fault line running through the Ombretta Valley. The signature of glacial landform shaping is clearly evident in the region (Carton et al., 2017). Moreover, the landscape bears abundant evidence of past glacial activity, with well-preserved, although small, moraine ridges, roches moutonnées, hanging valleys, and glacial cirques.

## 2.2 Climate

The climate over the Marmolada massif exhibits pronounced variations with elevation, reflecting the steep altitudinal gradients typical of high mountain environments. At mid-elevations—around 1500 m—the climate is classified as subalpine/subarctic (Köppen Dfc). In these zones, winters are long and cold, with average temperatures near –4 °C, while summers are brief and mild, averaging between +13 and +15 °C; extreme summer temperatures can reach up to 28 °C, and winter lows may drop to –15 °C. As elevation increases, temperatures decline significantly. At approximately 2100 m the mean annual temperature is around 3.6 °C, and by 3000 m it falls to roughly –2 °C. Within the alpine tundra zone—typically spanning 3000 to 3300 m (Köppen ET)—even the warmest month barely exceeds the freezing point, with July temperatures ranging from +2 to +4 °C and record winter lows nearing –21 °C, as observed at Punta Rocca (3250 m) in January 2020. The slope exposed to the north is partially in permafrost conditions (Boeckli et al., 2012). Precipitation also varies with altitude, increasing from about 1000 mm at lower elevations to over 1500 mm above 2500 m. The surrounding valleys typically experience 60–70 days per year with measurable precipitation (≥1 mm), with a pronounced summer maximum driven by convective storms and frontal systems.

## 2.3 The glacier

The glacier exhibits a well-documented history of retreat, particularly accelerating in recent decades following the conclusion of the Little Ice Age – LIA (Crepaz et al., 2013; Santin et al., 2019). This retreat has resulted in the fragmentation of the glacier into distinct sectors, with the Main Glacier (ID 941: CGI-CNR, 1962) representing the central unit. The Punta Penìa Glacier (ID 941.2) flanks the massif's highest peak to the north, while the Central Glacier (ID 941.1) lies below Punta Penìa.

A comparison of glacier inventories highlights the dynamic nature of the Marmolada glacial system. The 1962 inventory (CGI, 1962) identified up to eight glaciers within the massif. Subsequent years, however, witnessed significant changes, including the disappearance of some glaciers and the fragmentation of others. The most recent inventory (Smiraglia and Diolaiuti, 2016) streamlined the number of glacial units to seven and implemented revisions to their boundaries. The collective glacial front currently extends nearly 3 km.

It is worth highlighting that during World War I (WWI) the Alpine glaciers offered sufficient thickness to accommodate tunnels and subterranean defences, with the Marmolada Glacier housing about 300 Austro-Hungarian soldiers (Bondesan et al., 2015). By 2004, the maximum glacier thickness had already diminished to around 45 m, while other glaciers experienced even more drastic reductions (Pasta et al., 2005).

## 2.4 The collapse

The collapse in the early afternoon of 3 July 2022 occurred at an altitude of approximately 3200 m and affected a small, isolated glacier or glacieret located in a small cirque just below Punta Rocca (3309 m asl). This massive ice body was about

160 m wide, 140 m long and 30° steep when it collapsed. The failure occurred along a large median traverse crevasse and the detachment area was shaped like a rectangle of 70 m by 90 m leaving an overhanging ice wall of about 23 m (Bondesan and Francese, 2023). The failed mass ran over the normal route to the summit (Punta Penia – 3343 m asl), overwhelming many of the mountaineers climbing at that time and resulting in 11 people killed and seven severely injured. The avalanche reached a maximum speed of 80-90 km/h and travelled 2.3 km (Fig. 2) before ending in a lateral valley at an elevation of 2330 m asl.

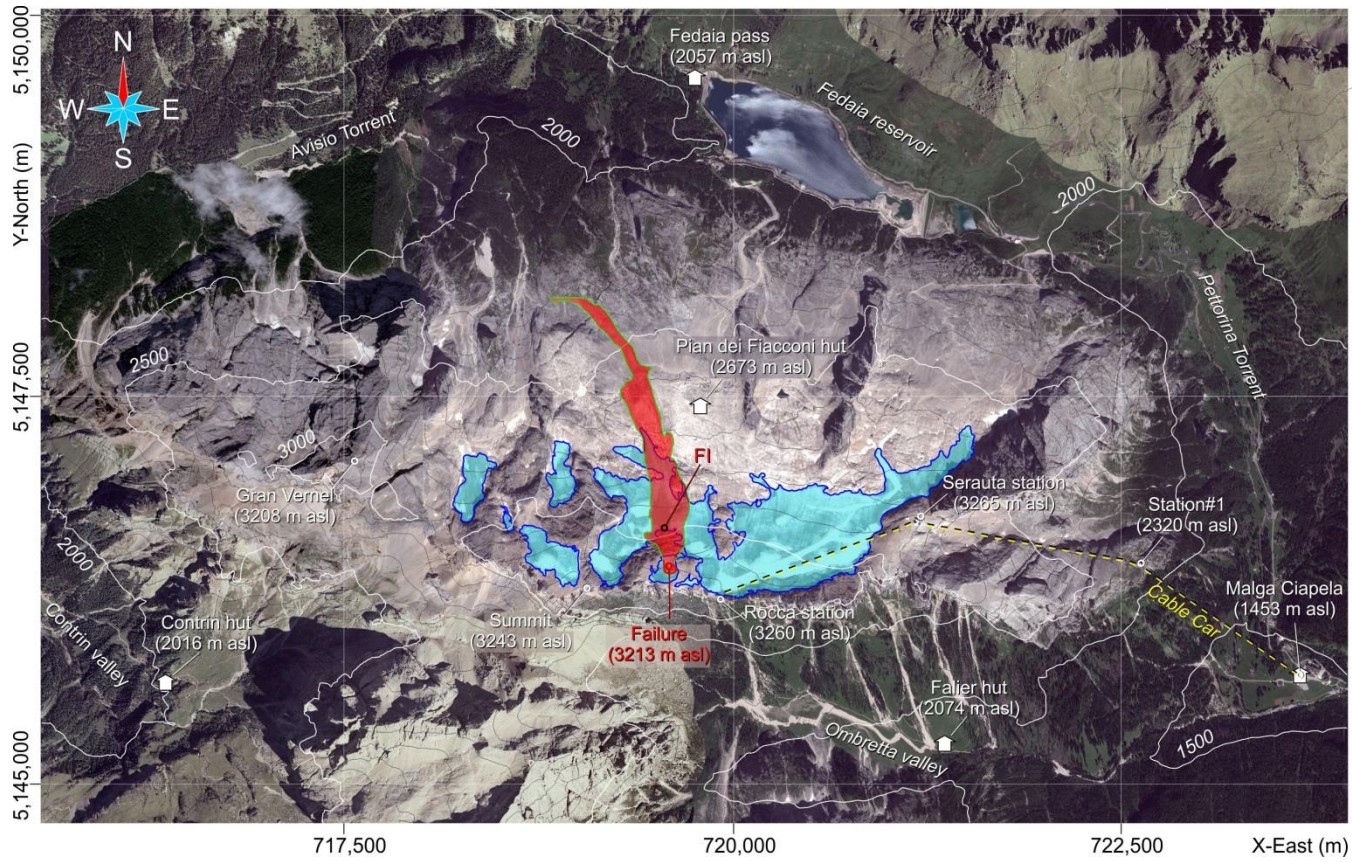

**Figure 2: Satellite imagery of the Marmolada massif showing the flow path of the ice avalanche which extended from 3213 m asl to 2300 m asl. FI indicates the position of the first vertical impact of the collapsed mass on the lower glacier surface. The ice avalanche was only 250 meters away from the hut "Pian dei Fiacconi", which was already destroyed by a snow avalanche in 2020. Aerial and satellite imagery source: Pléiades Neo, AIRBUS Defence & Space; Autonomous Province of Trento (PAT).**

At the time of the collapse, crevasses were completely water-filled and approximately 11,000 $m^3$ of water were suddenly discharged with the failed mass (Bondesan and Francese, 2023). The flowing mass along its travel path discharged the subglacial till and also the loose glacial debris that had been deposited at the front in recent decades. A cloud of droplets, fine ice crystals, ice fragments, and debris, forming the flowing mass, widened over a front larger than 500 m and covered a

surface area of approximately 350,000 m². The event was recorded by the seismic network and the resulting magnitude was estimated as 0.6 ±0.3.

## 3 Data, methods and concept

The multi-disciplinary approach of the present study involves several research topics and different data types along with the specific methods of data acquisition, analysis and processing. Data were gathered in a large database comprising a wide range of glaciological and meteorological records, along with historical and modern topographic maps, aerial and satellite imagery, geophysical data, geological and geomorphological data collected and catalogued over the past two decades (see Supplementary Material for details).

### 3.1 Historical glacier evolution and recent meteorological and glaciological data

Time-lapse reconstruction of the glacier geometry (1880-2021) was a two-step process. Initially, the bedrock topography was determined using RES and laser measurements, where it was covered by ice and exposed, respectively. Subsequently, the glacier surface was reconstructed at different time periods using digitization techniques applied to historical maps, digital maps, aerial and satellite images, and, for the most recent periods, light detection and ranging (LIDAR) and terrestrial laser scan (TLS) data. The top (ice surface) and bottom (bedrock surface) were computed using aligned 5 m × 5 m grids, ensuring that both surfaces share the same node coordinates to simplify calculations. A total of 12 triplets "area, surface-area, volume" could be calculated from the set of available data. Volume error estimates range from a maximum of 25% in the year 1880 to a minimum of 7% in the year 2021. A comprehensive summary of the used data is listed in the Supplementary Material along with the computational error theory.

Available meteorological data were processed to compute yearly and monthly averages for the reference period 1900-2020 and for the year 2022. Nine weather stations located around the Marmolada massif (see Supplementary Material for the location map) were considered to provide daily time series of air temperature (min, avg. and max), snow thickness and precipitation. Among these stations, Punta Rocca (PRC) is the closest to the failure site as it is located at 3200 m asl at a distance of approximately 400 m. Missing values were reconstructed (see Supplementary Material for details) using a variety of techniques (Bondesan and Francese, 2023). The periodical observations on the glacier front, carried out by the Italian Glaciological Committee (IGC), were also considered in the form of yearly time series to further estimate retreat rates for the period 1985-2023.

### 3.2 Permafrost and ice temperature data

The permafrost information for the study area was inferred from the 25 m x 25 m model for the Alpine chain (Boeckli et al., 2012; Gruber, 2012) and from the nearby borehole of the Piz Boè (PZB) equipped with operational temperature sensors since the year 2010 (Crepaz et al., 2011).

The model devised by Boeckli (2012) is based on Mean Annual Air Temperature (MAAT) of the period 1960-1990. PZB data confirm this model and provide some information about ongoing permafrost warming in the region. The PZB borehole is located at 2905 m asl, approximately 300 m lower than the failure site, on the eastern slope of the Piz Boè massif, about 8 km to the NNW.

Ice temperature estimates for the Marmolada glacier were inferred by using and updating published data (Haeberli et al., 2004a; Fischer et al., 2022). In early August 2024, a borehole was drilled in the residual ice body just above the failure scarp using a steam-based drill bit and was equipped with four temperature sensors. The sensors were located just above the ice-bedrock interface (at a depth of 11.5 m below the surface) and at -2.5 m, -5.5 m, -8.5 m from the ice surface respectively.

### 3.3 Satellite imagery

Pre- and post-event satellite images were taken by the Pléiades Neo constellation (AIRBUS Defence & Space),
pansharpened, pancromatic 30 cm native Ground Sample Distance (GSD); 6 multi-spectral channels, 1.2 m native GSD. Pancromatic sensors span from ~450 nm to ~800 nm; multispectral sensors span from ~380 nm to ~880 nm. A pre-failure shot was taken on Jun 20, 2022, h 10:06:67 GMT while two post-failure shots were taken on Jul 8, 2022, h 10:02:07 GMT and Jul 9, 2022, h 10:20:21 GMT respectively.

The resolution of each normalized band was increased by a factor 3 to enhance readability on a 4k display; a Lanczos-3
interpolation kernel was used, since it provides a good balance between visual image quality and the introduction of undesired spectral components (Madhukar et al., 2013). The high frequency content of each normalized band was then enhanced by subtracting from the data the output of a linear Gaussian filter having standard deviation equal to 0.9 and a gain factor of 2. The Normalized Difference Water Index (NDWI) was finally calculated computing the difference between the NIR (Near InfraRed) and green bands (McFeeters, 1996).

### 3.4 Seismology

The seismicity of the area was inferred analysing data provided by the NOAN - North-Eastern Alps Network of OGS (CRS, 2024; Rebez et al., 2024). The seismic stations are located south-east and north-west of the glacier.

The failure-generated earthquake was located using an automatic routine based on the HYPO71 algorithm (Lee and Lahr, 1975) that mostly uses logic and arithmetic resulting in high computational efficiency. Additional information was retrieved
via the filter picker algorithm (Lomax et al., 2012). Event onset detection was later refined via a detailed picking of the body wave phases (in this case P-wave only) in the records of six of the surroundings NOAN stations. The localization of the event appears accurate as it corresponds to the real position with an error of about 2km despite there being no NOAN stations in the north-east quadrant.

### 3.5 Geometry of the failure zone

At the collapse site, surface morphology was reconstructed immediately after the event mostly using UAV (Unmanned Aerial Vehicle) LIDAR and TLS data while the buried bedrock geometry was reconstructed using a combination of pre- and post-failure geophysical profiles. The residual ice surface and the outcropping bedrock were modelled using a 0.05 m by 0.05 m mesh. Post-failure RES profiles were collected in October 2022 just above and around the failure edge. The profiles were collected using a Subsurface Interface Radar (SIR) 4000 equipped with 200 MHz and 500 MHz antennas. Some additional scanlines, located around the failure edge, were recovered from the 2004 RES campaign (Pasta, 2004). These data required to be converted back to amplitude time series (see Supplementary Material for details) prior to be geo-referenced and re-processed. Seismic Unix (Stockwell; 1999; Picotti et al., 2017) was utilized for the purpose. The buried bedrock response was then interpolated over a 1.0 m grid. This set of RES profiles was used to reconstruct the geometry and physical properties of the ice-rock interface beneath the failed ice mass.

### 3.6 Glacieret stability and back analysis

In a simplified way, disregarding the progressive failure, the stability of the glacieret was assessed along a representative 2D cross section by means of the Limit Equilibrium Method (LEM), which is routinely used for slope stability analyses in geotechnical engineering. LEM is based on the principle that a rigid mass (in this case made by ice), will fail when the driving forces, due to gravity and external loads, exceed the resisting forces, due to shear strength along a defined failure surface. In the present study, the main driving force is the weight of the unstable mass, but further destabilising actions, represented by hydrostatic forces in various configurations, were considered, as will be explained later. The calculated driving actions are compared to the available resistance, which is calculated according to Mohr-Coulomb's shear strength criterion. Referring to this shear strength criterion, specific strength parameters are assumed for the involved materials, i.e. ice and rock. The method considers the equilibrium of forces and/or moments along a predefined failure surface. The stability is expressed through the Factor of Safety (FoS), which is the ratio between resisting and driving actions, assuming that the mass is in a condition of incipient motion. The lower the FoS the higher the possibility of instability and collapse. If FoS is less than one, the slope is unstable and FoS = 1 is assumed as the limit stability value.

In the framework of LEMs, slice methods are used to analyse the stability of slopes by dividing the unstable mass into vertical slices. These methods evaluate the equilibrium of each slice while considering forces acting within and between them. Each slice is analysed separately, considering its weight, normal force, shear force, and inter-slice forces. The stability of the entire slope is determined by calculating FoS based on shear strength and equilibrium conditions. Among the variety of methods of slices available to determine FoS, three different methods were used in this analysis, namely, Janbu simplified - Js (Janbu et al., 1956), Janbu corrected - Jc (Janbu, 1954; 1973), and GLE/MP (General Limit Equilibrium; Morgenstern and Price, 1965), by using the SLIDE2 software from Rocscience®. In particular, the Janbu's simplified method is based on equilibrium of slices along two orthogonal directions. It assumes only horizontal interslice forces and thus the interslice

shear forces are zero (Janbu et al., 1956). The Janbu corrected FoS is obtained by multiplying the Janbu simplified FoS by a modification factor, which is function of the slope geometry and the strength parameters of the material. The Janbu modification factor is an attempt to compensate for the fact that the Janbu simplified method satisfies only force equilibrium and assumes zero interslice shear forces. Instead, GLE/Morgenstern-Price method assumes a half sine function to define

interslice forces and it is based on a complete equilibrium of slices along two orthogonal directions and with respect to rotation. This is why this method is 'rigorous'. It should be noted that, although the failure surface was fully known, in the LEM model it was defined only at its base (i.e., at the ice–rock interface), with the aim of comparing the predicted and observed failure scarp within the ice mass.

The parameters considered for the stability analysis are cohesion and friction angle, defining the shear strength of the glacial

mass, and the underlying bedrock substrate.

Additional and more detailed information on data and methods can be found in the Supplementary Material and in Bondesan and Francese (2023) in both the paper and the appendix.

## 4. Results

### 4.1 Evolution of the Marmolada glacier

Field measurements available since the early Eighties (Fig. 3A and Fig. 3B) clearly show the progress of the retreat. Distance and elevation measurements are referred to specific benchmarks located in the western, central and eastern sector respectively. Retreat in the eastern sector is not as drastic as in the central and western sectors. The averaged ~30-year trend clearly exhibits three different gradients for both distance and elevation. The sudden jump occurring in the year 2000 is caused by a partition of the glacier in correspondence with a vertical step in the outcropping bedrock. The three gradients

could be also recognized in the graph reporting the lowering of the glacieret's surface (Fig. 3C) right at the failure site.

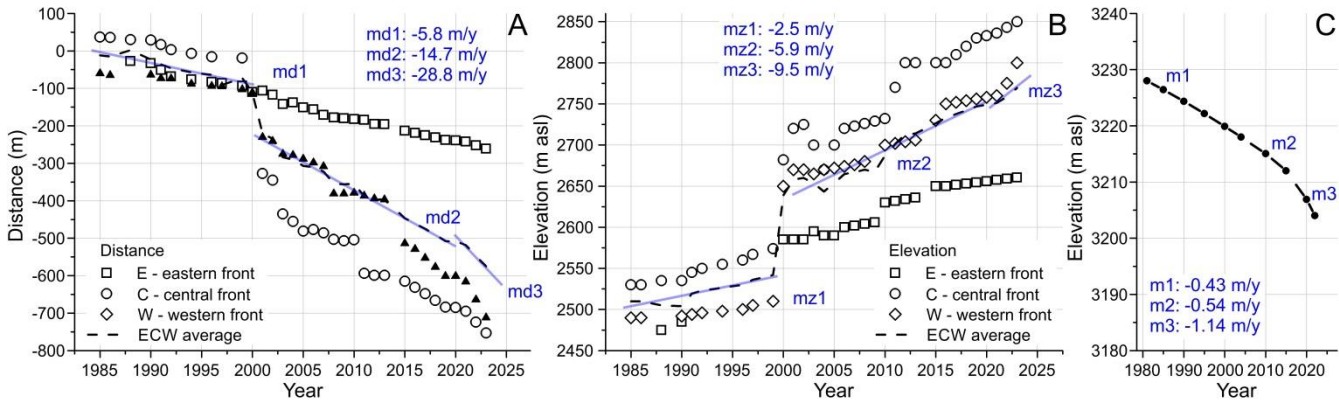

**Figure 3: Measurements of front migration and surface lowering for the Marmolada Glacier. A reference point was available for each of the different glacier sections (see also Fig. SM 4). (A) Inclined distance taken from the glacier front over the period 1985-2023. (B) Vertical distance taken from the glacier front over the period 1985-2023. (C) Progressive lowering of the glacier surface**

**in the period 1980-2022 computed right at the point of collapse.**

Area, surface and volume, computed for 12 time frames since 1880, show the longer-term retreat trend (Fig. 4A, Fig. 4B). The space migration of the centroid of the glacial front is also a good indicator of the retreat trend (Fig. 4C). After approximately one century (1982) the surface area was 53.9±0.7% of the initial surface, while the volume was 26.2±12.5%

of the initial volume. In the year before the failure (2021) the surface was 26.4±0.5% of the initial surface, while the volume was 10.2±7.5% of the initial volume. In a period of about 130 years the Marmolada glacier lost approximately 73% of the surface area and 90% of the volume.

This evolution of the Marmolada glacier, in terms of surface reduction and mass balance, shows a trend similar to that of the other European glaciers (Bondesan and Francese, 2023) and glaciers worldwide (Dyurgerov and Meier, 2004; Hugonnet et

al., 2021).

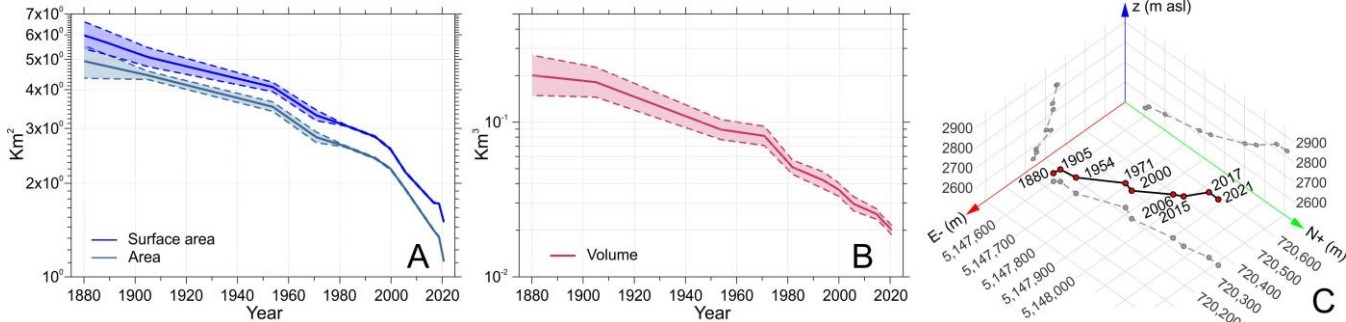

**Figure 4: Over a century (1880–2021) of Marmolada Glacier retreat in terms of area, surface area (two-dimensional measure of the slope's terrain surface, accounting for all inclinations and undulations), volume, and front migration, obtained through GIS/CADD analysis and processing. (A) Area and surface area reduction, the interval of confidence is represented by dashed lines.**

**(B) Volume reduction, the interval of confidence is represented by dashed lines. (C) Tri-dimensional spatial migration of the centroid of the glacier front (the centroid was calculated as the arithmetic mean of the xyz coordinates of the nodes in the polyline representing the glacial front in different years, see SM in Bondesan and Francese, 2023 for details).**

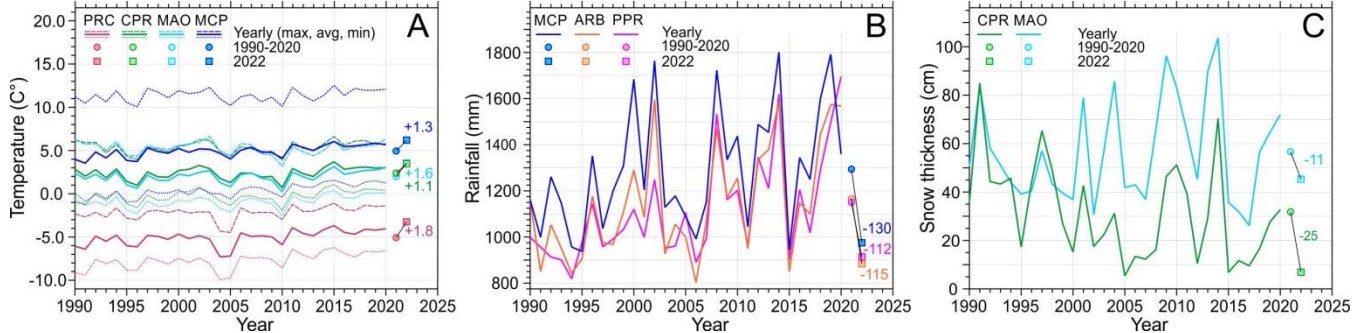

**Figure 5: Annual meteorological variables pertaining to the ~30-year reference period 1990-2020, see Supplementary Material for station code and position: PRC is located at 3250 m asl and it is the closest to the failure site (~400 m); in each panel is graphed: the annual curve; the ~30-year average (circle mark); the 2022 value (square mark). The numbers on the right of each panel represent the Δ between the ~30-year average and the 2022 value. (A) Minimum, maximum and average temperature curves, Δ were computed for the average values only. (B) Rain/snowfall. (C) Snow thickness. See text for further details.**

## 4.2 Meteorological variables and permafrost conditions

The year 2022 proved to be particularly anomalous as for temperature, rainfall/snowfall and snow cover in comparison to the ~30-year period 1990-2020.

The annual temperatures for the year 2022 showed a marked positive shift in comparison with the 30-year average for the
period 1990-2020. The positive shift of average temperature (labeled avg in Fig. 5A) was +1.8°C, +1.1°C, +1.6°C and +1.3°C for the stations PRC, CPR, MAO and MCP respectively (Fig 5A). The positive shift for the station PRC is particularly high and it is also higher considering the positive shift of the minimum value that happened to be equal to +1.9°C. Contrarily annual precipitation for the year 2022 (Fig. 5B) showed a negative shift of about 120 mm (average of three stations) corresponding to a reduction in annual precipitation of approximately 22%. Finally considering the snow
thickness for the year 2022 (Fig. 5C), a negative shift of about 18 cm (average of two stations) was observed, corresponding to a reduction in annual snow thickness by about 41%.

A more accurate analysis could be done considering the monthly time series (Fig. 6). Data from the year 2022, similarly to yearly time series analysis, are compared to the 30-year period 1990-2020.

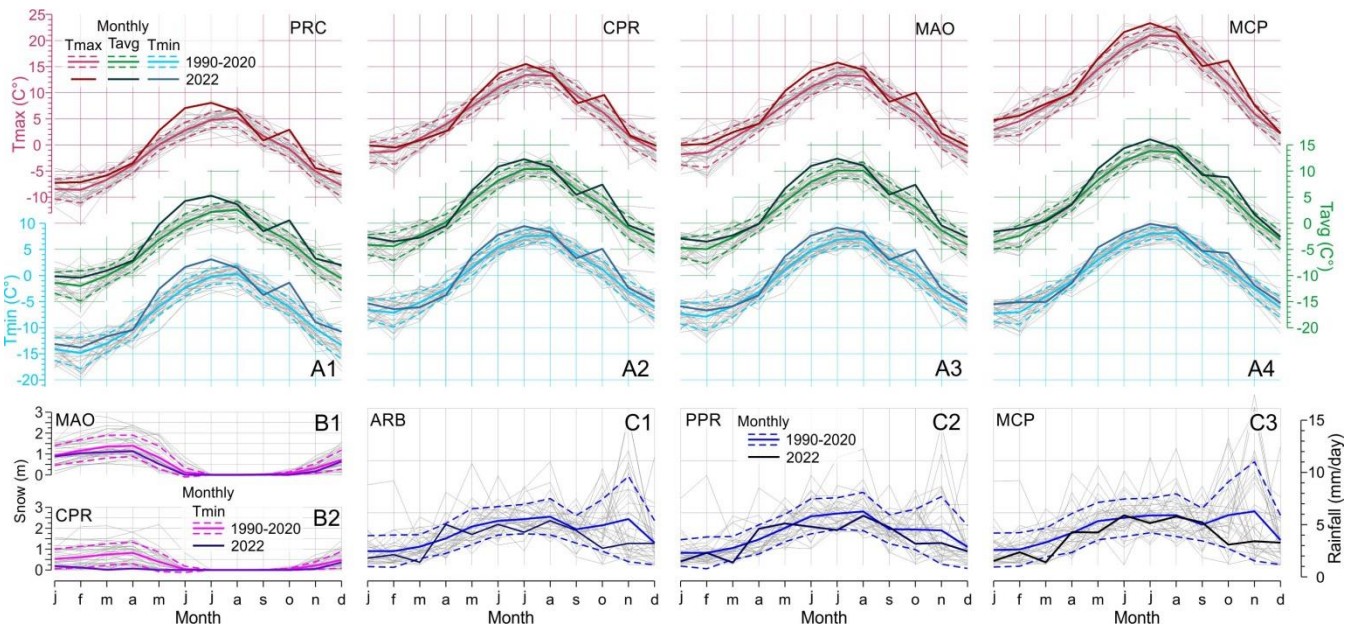

**Figure 6: Monthly meteorological variables related to the ~30-year reference period 1990-2020, see Supplementary Material for station code and position: PRC is located at 3250 m asl and it is the closest to the failure site (~400 m); in each panel is graphed: the monthly curve per each year (solid line in grey colour); the ~30-year average (solid line); the standard deviation interval (dashed line); the monthly curve for the year 2022. (A) Minimum, maximum and average temperature curves. (B) Snow thickness. (C) Rain/snowfall. See text for further details.**


Monthly temperature averages (Fig. 6A) clearly show how during the period May-July 2022 the recorded temperatures fall outside the standard deviation range for the 30-year reference period as for minimum, average and maximum values. It is worthwhile to notice that the values recorded at the PRC station, during the year 2022, reach the absolute maxima for the period May-July (Fig. 6A1). With regard to rainfall/snowfall (Fig. 6C) the 2022 values fall within the standard deviation range for the reference period although the curve is very close to the lower bound.

Snow cover during winter and spring of the year 2022 was particularly thin (Fig. 6B). The thickness value was just below the average (Fig. 6B1) north of the Marmolada glacier (MAO station) and approximately along the lower bound of the standard deviation interval (Fig. 6B2) south of the Marmolada (CPR station).

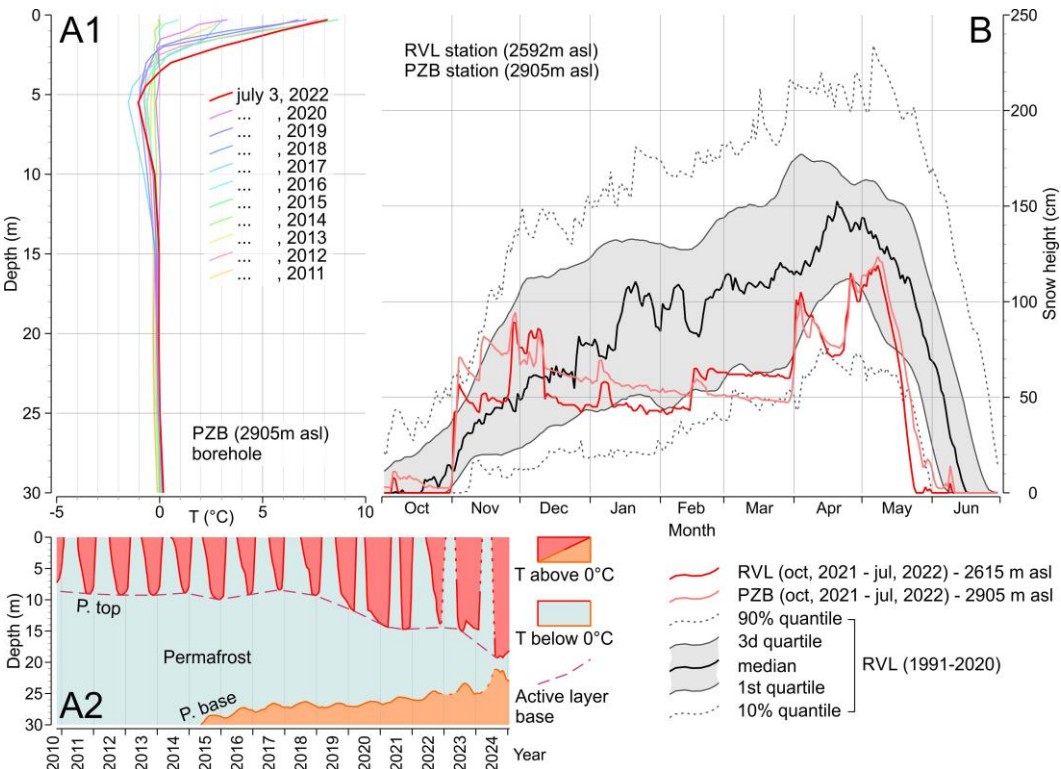

**Figure 7: (A1) Borehole temperature measurements collected on July 3 of each year since 2011 at the PZB station (see Supplementary Material for station code and position). (A2) Temporal evolution of the depths of the active layer base (P. top) and the permafrost base (P. base) from October 2010 to December 2024 at the PZB station. (B) Seasonal snow height at RVL and PZB stations in 2022 in comparison to the average snow height of the period 1990-2020 measured at the RVL station (see Supplementary Material for station code and position).**

Analysis of snow thickness was extended including daily time series from two additional stations (Fig. 7B) to overcome local variability uncertainty (exposition, wind, etc.). These stations are located a few kilometres north (PZB) and north-east (RVL) to the Marmolada glacier. Snow cover in the period October, 2021 – June, 2022 is compared to snow cover during the

30-year reference period 1991-2020 recorded at the RVL station. The snow-cover curves, in the first months of the year 2022 are way below the average and close to the 10% quantile limit for both the two stations. In early April and early May the curves enter the first quartile of the reference period but in late May the curves drop to almost zero thickness falling below the 10% quantile. These data show clearly how in the year 2022 the snow melted one month early and that at an elevation of 3000 m asl there is very little snow left after mid-June. The negative shift of the RVL curve depends mostly on

its lower elevation.

Permafrost occurs around the detachment zone and in ice-free terrain. It was most probably colder and reached lower altitudes during the LIA. Due to ongoing global warming, permafrost rocks are warming rapidly (Biskaborn et al., 2019, Noetzli et al. 2024). The PZB borehole temperature at a warm, east-exposed site documents a thick active layer and a nearly-isothermal permafrost layer with temperatures ranging between -1.5°C and 0°C in 2022 (Fig. 7A1). Moreover, the

temperature starts to increase above 0°C at about 25 m depth (Fig. 7A1), marking the base of permafrost. Despite the uncertainty about the permafrost conditions at the Marmolada detachment site, we estimate the near surface temperature to be colder than at PBZ (at about -2° C), due to its northerly exposure.

Analysis of the temperature time series from the PZB borehole (Fig. 7A2) reveals that the base of the permafrost has risen from approximately 30 m below the surface in 2014–2015 to around 22 m below the surface in 2024.

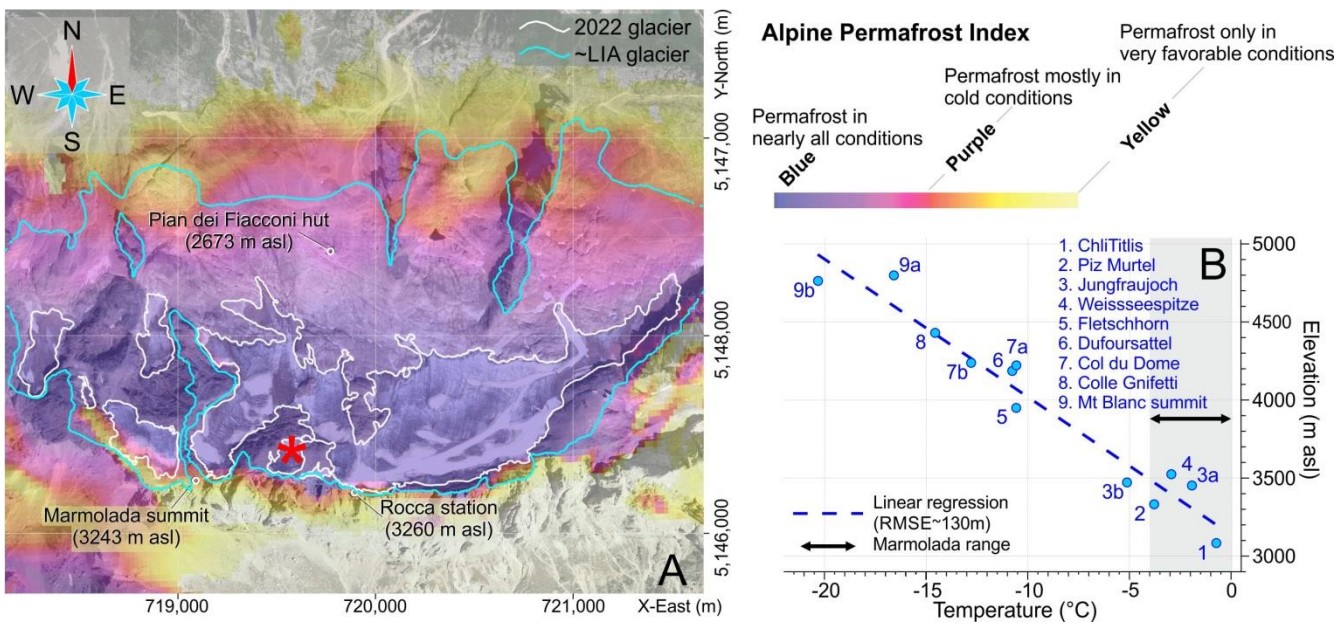


Figure 8: (A) Modelled permafrost occurrence in the Marmolada massif (after Boeckli et al., 2012). The detachment zone is marked with a red asterisk. A strong contrast exists between the sun-exposed, ice-free south wall with its warm, probably non-frozen rocks, and the cold north-slope oriented away from the sun and with widespread permafrost. The glacier boundary in July 2022 is outlined with a white line while the line in cyan colour outlines the boundary of the LIA glacier. (B) Ice temperature
measured at different sites in the Alps (redrawn from Haeberli et al., 2004a and updated including the Weissseespitze from Fischer et al., 2022). The thermal conditions at the Marmolada detachment fit the linear regression curve. The large grey band for the Marmolada detachment site indicates the uncertainty in temperature information obtained from different times and using different methods/accuracies.

Modelled permafrost occurrence in the Marmolada massif (Fig. 8A) shows that, at the failure site (as of 2012), permafrost is present under nearly all conditions (Boeckli et al., 2012). Ice temperature, at the failure site (Fig. 8B), could similarly be estimated in the range of 0°C to -4°C, based on a simple 'T-Elevation' function developed for the Alpine region (Haeberli et al., 2004a; Fischer et al., 2022).

## 4.3 Characterization of the failure zone

Pre- and post-failure satellite imagery provided some insight on the collapsed glacieret. Image analysis mostly focused on the failure surface and on the water content.

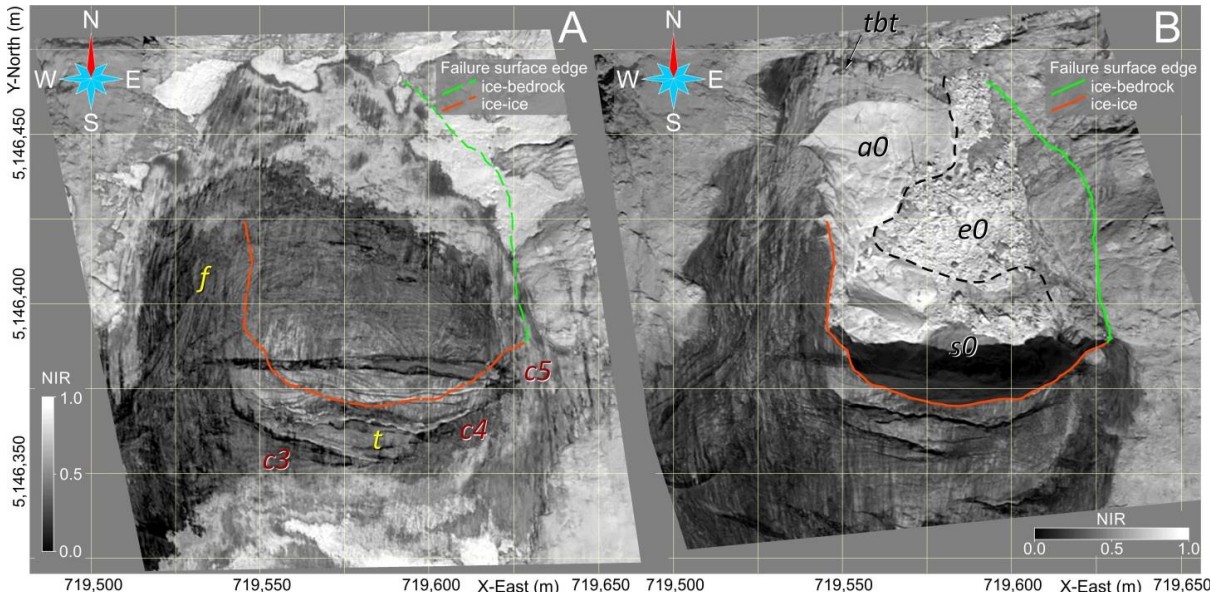

**Figure 9: Near-InfraRed satellite images (Pléiades Neo, AIRBUS Defence & Space) of the glacieret taken before and after the failure. (A) Date of acquisition: June 20, 2022; f –foliated ice; t – small supraglacial torrent flowing eastward across the glacier; cx - crevasse. (B) Date of acquisition: July 8 and July 9, 2022; a0 – relatively thick residual ice layer; s0 – regelation cryoconite-enriched thin ice layer; e0 – exposed bedrock; tbt – thin basal till. See text for details.**

The NIR (Near-InfraRed) map (Fig. 9A) before the failure clearly shows a highly foliated ice body with a wedge-shaped median/traverse crevasse (c5 in Fig. 9A) along with a series of the rear crevasses. The median crevasse appears to be filled with water with floating ice and snow. The maximum width of the median crevasse occurs on the eastern side and it measures approximately 5 m.

A small supraglacial torrent (marked with letter 't'), fed by meltwater, flows eastward over the glacier's surface. It has formed atop one of the rear crevasses (c4 in Fig. 9A) and is oriented almost transversely to the slope, with its discharge occurring right at the intersection with the large median crevasse. The flow is transverse because the lower face of the crevasse forms a counter-slope, preventing the water from flowing in the longitudinal direction.

The same map after the failure (Fig. 9B) provides important details on the detachment surface. The area of the basal detachment surface is about 6200 m$^2$ and the rupture occurred at the boundary of three different materials: ~32% along a foliation (a0, ice-ice); ~36% almost at bedrock (s0, ice-re-gelation thin ice); ~32% at the top of the bedrock (e0, ice-bedrock). The bedrock in the e0 sector is just partly visible because it is completely covered by ice/snow blocks and ice debris.


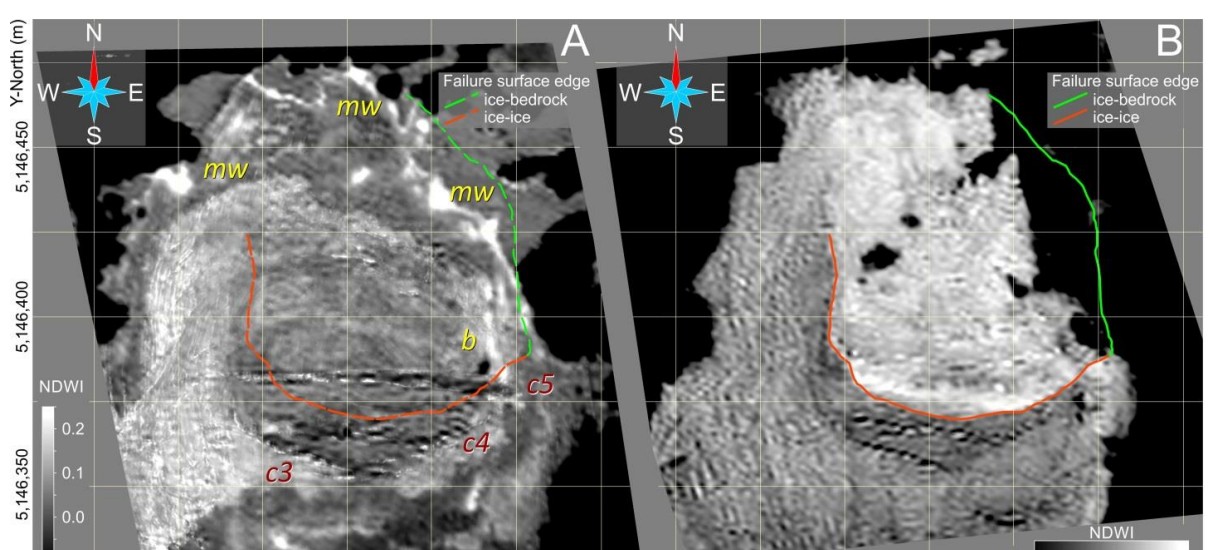

**Figure 10: NDWI (Normalized Difference Water Index) map obtained via processing of multispectral satellite images (Pléiades Neo, AIRBUS Defence & Space) of the glacieret before and after the failure. (A) Date of acquisition: June 20, 2022; b – longitudinal bédière; mw - meltwater. (B) Date of acquisition: June 20, 2022. See text for details.**


The NDWI (Normalized Difference Water Index) map (Fig. 10A) before the failure is mostly effective in detecting the presence of water (outlined by high values of NDWI). A bédière-like outflow pattern is visible in correspondence to the eastern edge of the large median crevasse (marked by letter b in Fig. 10A) indicating that the crevasse is mostly water-filled below a snow/ice plug. A high NDWI response is also visible along the edge of the various crevasses suggesting the

presence of water below plugs of snow and ice.

Meltwater is finally visible along the snow-ice boundary and it probably accumulates because of the counter slope caused by the snow layer. Apparently there is no water discharge at the toe of the glacieret. Very little evidence of water is visible after the failure (Fig. 10B) indicating that most of the englacial water was discharged during the collapse.

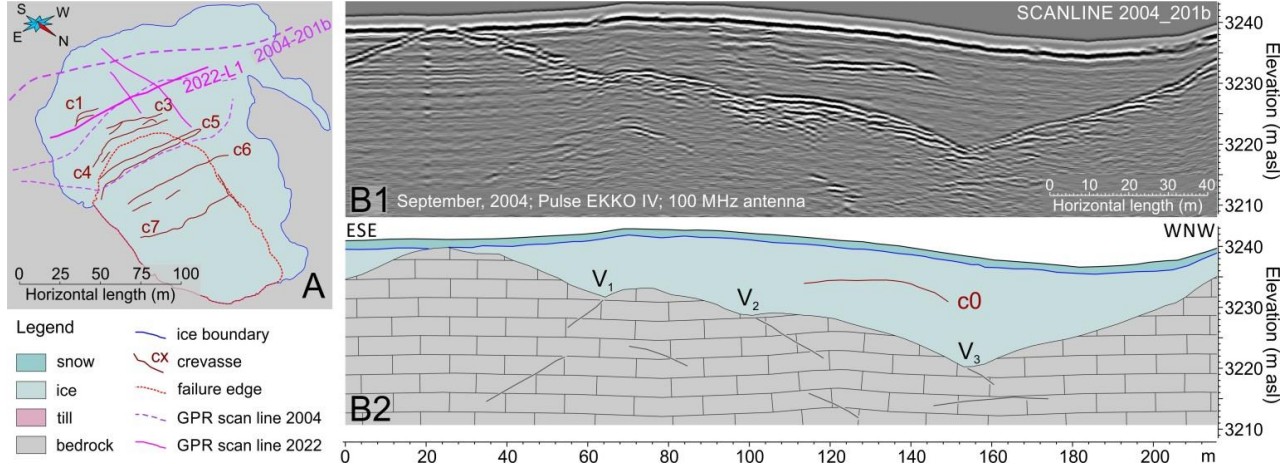

**Figure 11: RES profile collected just above the failure scarp before the collapse. (A) Sketch map showing the location of the profiles. (B) Scanline 2004-201b (reprocessed from Pasta, 2004): data (B1) and interpretation (B2); bedrock and crevasse response is clearly visible.**

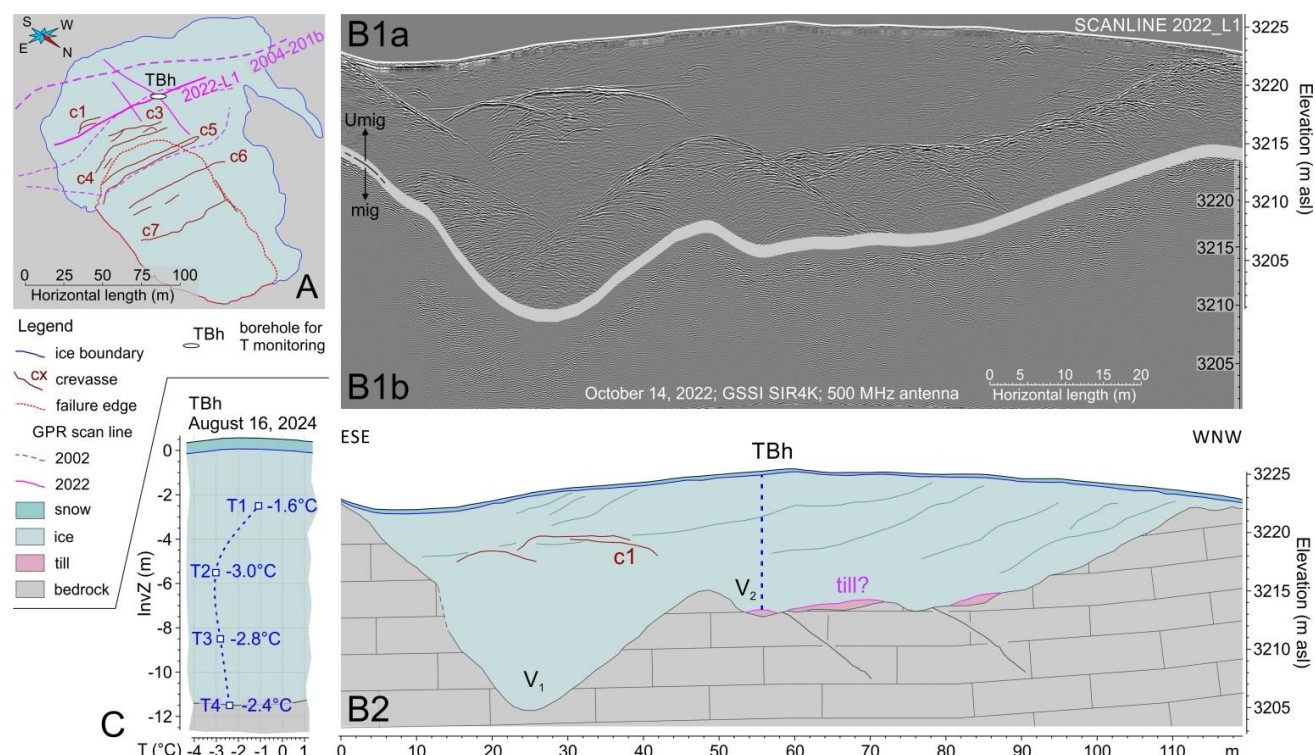

**Figure 12: RES profile collected just above the failure scarp after the collapse. (A) Sketch map showing the location of the profiles. (B) Scanline 2022-L1: (B1a) Processed data, not migrated; (B1b) Processed and migrated data with a focus on the bedrock response; the maximum spatial shift of the bedrock in the migrated data was less than 1 m. (B2) Interpretation; bedrock morphology is clearly imaged along with crevasses, foliations and some thin layers of anomalous response interpreted as basal tills. Troughs Vi could be correlated across the two different profiles. (C) Temperature record from the borehole measurements taken on August 16, 2024. See text for further details.**

RES profiles (Fig. 11 and Fig. 12) aided in outlining the buried bedrock morphology and the ice structure above and on the side of the failure scarp. A first profile collected several years before the failure (Fig. 11B) outlines the bedrock morphology in the upper slope, at a distance of approximately 50 m from the failure scarp. The bedrock exhibits three incisions (V1, V2 and V3) with a maximum ice thickness of about 20 m at incision V3. The response of a crevasse (c0), no longer present, is visible in-between V2 and V3. A second profile collected three months after the failure just above the failure scarp (Fig. 12B) shows similar features but with greater resolution. Two incisions are clearly visible and they nicely correlate with V1 and V2 of the previous profile while V3 is just lightly engraved. High-amplitude diffraction hyperbolas mark the top of some humps in bedrock morphology.

The maximum ice thickness is about 16 m at incision V1. Diffractions and reflections from crevasse c1 are visible at the beginning of the radar profile. A series of low-amplitude and east-dipping reflectors are embedded in the ice body. It is the response of the ice foliations described in the satellite images. Very short wavelength radio waves are needed to outline these tiny reflectors. Gently east-dipping bedding planes are partly outlined by reflections within the bedrock and the response of some fractures is also clearly visible. The profile was collected almost parallel to the strike of the strata.

Ice foliations and rock beddings are barely detectable in the 2004 profile due to the low frequency of the radar transducer (100 MHz) while foliations are clearly imaged in the RES-scan collected in October 2022 (Fig. 12B). The foliation planes apparently dip eastward, with an inclination of less than 6-7 degrees. Some low-reflectivity thin layers are visible right on top of the bedrock in the central and western segment of the 2022 RES-scan. The poor degree of reflectivity suggests the presence of a dispersive medium, typical of cohesive sediments, which could be reasonably interpreted as a basal till comprised of finely crushed rocks.

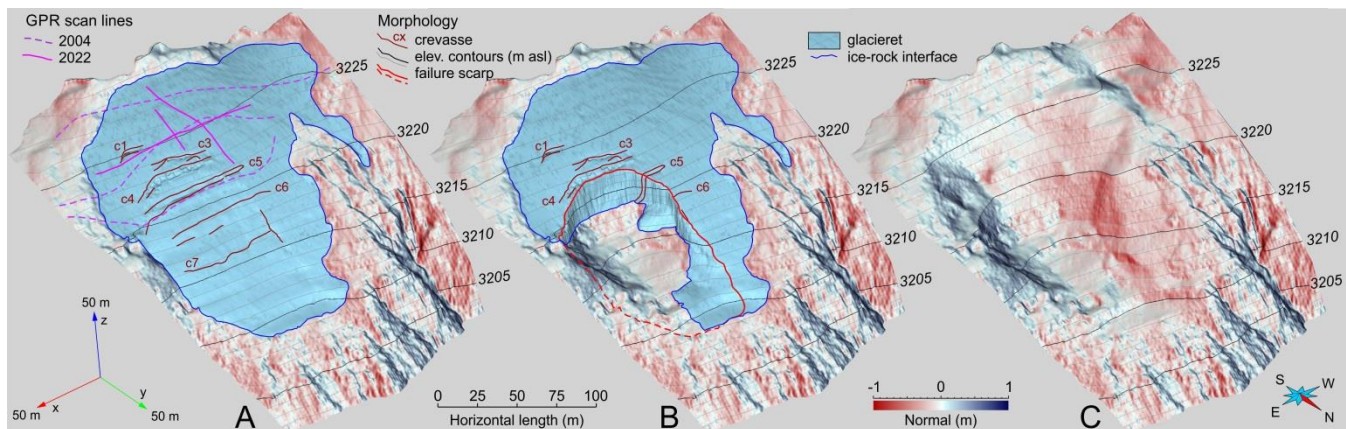

**Figure 13: 3D-morphology of the rock-ice ensemble modelled right at the detachment site. (A) Pre-failure settings; the volume of the glacieret is approximately 159,000 m³. (B) Immediate post-failure settings; the residual ice volume is approximately 89,000 m³. (C) Bedrock morphology reconstructed via RES imaging below the residual ice body; the bedrock niche where the glacier failed had a basal dip that ranged from 41 degrees in the rear portion to 20 degrees in the frontal part.**

The reconstruction of the buried bedrock morphology allowed for a precise modelling of the area of the collapse (Fig. 13). The volume of the glacieret, prior to collapse, was approximately 159,000 m$^3$ (Fig. 13A), contained within a perimeter of ~800 m, while the collapse involved about 70,000 m$^3$ (Fig. 13B) of ice, water and various debris leaving ~89,000 m$^3$ of ice in-situ. The perimeter of the failed mass is about 375 m and the failure surface, including the nearly vertical faces, was estimated at 9,700 m$^2$. The failed mass was hosted in a rock niche with a steep rear portion and a gentle frontal part. The bedrock inclination is approximately 41° in the rear portion and it is less than 20° in the frontal part (Fig. 13C).

The temperature of the ice surface along the failure scarp was measured several times with IR airborne sensors: the early morning after the failure (7:00 AM of Jul 4, 2022; at the beginning of September (Sep 7, 2022) and in mid-October (9:00 AM of Oct 14. 2022). The maximum temperature value was -9°C in summer time and -10°C in early autumn (See Supplementary Material for details and figures). These temperatures, always taken in the early morning, represent skin temperatures, but they are significantly lower than the nighttime air temperatures. The values are relevant because they have remained consistent and comparable over time. Reliable quantitative interpretation requires complex calibration of the sensor along with specific processing. Corrections on the order of several degrees are often necessary (Aubry-Wake et al., 2015) especially in the case of rough surfaces. Despite potential biases of several degrees, these measurements still indicate —at least qualitatively—that the ice body is unquestionably cold, with temperatures several degrees below freezing. The post-failure exposed bedrock temperature was similarly measured in the early morning on the day after the failure (M. Zumiani, personal communication) and ranged between 0 and -1°C. This skin value, equally influenced by nighttime air temperatures, indicates that surface thermal conditions are close to the melting point.

Following the drilling and sensor installation in August 2024, the vertical ice temperature profile was obtained two weeks later, after the thermal conditions around the borehole in the residual glacier had stabilized. The temperature at the ice-bedrock interface (-11.5 m below the ice surface) was -2.4 °C while the temperature within the ice body was of -2.8 °C, -3.1°C and -1.6°C at -8.5 m, -5.5 m and -2.5 m below the ice surface respectively (Fig. 12C). The average ice temperature can thus be estimated at some -2.4°C.

### 4.4 Seismology

Earthquakes in this area mainly occur far from the Marmolada massif (Fig. 14).

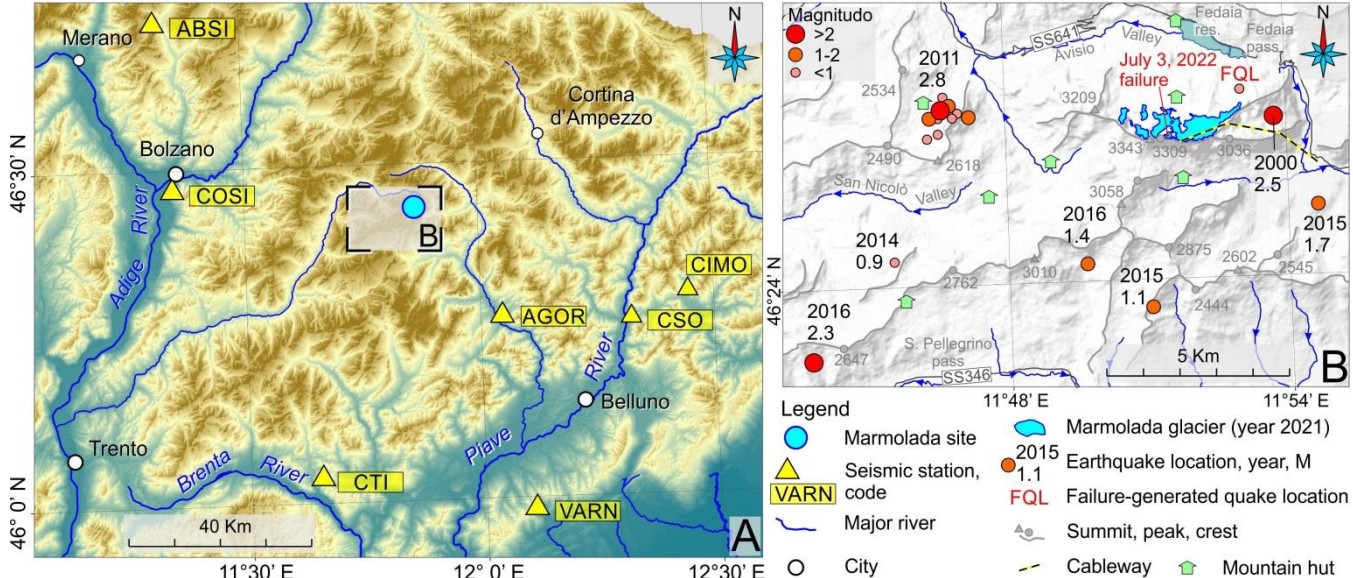

**Figure 14: Seismological settings. (A) Topographic map showing the location of the seismic stations around the Marmolada glacier. (B) Historical earthquakes occurred in the vicinity of the failure site; the FQL letter designates where the earthquake that was caused by failure was recorded by the nearby seismic stations; an earthquake with magnitude 2.5 was detected in the year 2000 just east of the glacier. On the day of the collapse only three earthquakes (Table 2) occurred in northern Italy and they were located more than 50 km from the glacier.**

Typical earthquakes occur east and southeast of Agordo and of the Piave river along the outer front of the Southeastern Alps (Gentili et al., 2011; Sugan and Peruzza, 2011). In the last twelve years, the seismicity within 5 km of Marmolada (Fig. 14B) has been limited to a cluster located south of Canazei (west of Marmolada) and a few M<2.5 magnitude events, including the M2.5 earthquake occurred in 2000, which is the closest event to Marmolada. The latter is located just south of Malga Ciapela. This uncertainty falls within the typical errors for seismic event location (D'Alessandro et al., 2011). An accurate positioning was then achieved minimizing the misfit of the arrival times and considering a regional velocity model devised for this area of the north-east Alpine chain.

The seismic motion generated by the ice avalanche was recorded in all three xyz components (Fig. 15). The period of greatest vibration intensity lasts about 60 seconds and approximately corresponds to the paroxysmal phase of the collapse, as can be inferred from the various available video footages. The seismic record (Fig. 15A1–A3) appears to be mostly dominated by horizontal components of body and surface waves (Rayleigh and Love). The strongest vibrations are visible about 20-25 seconds after the start of the wave train related to the event. The initial peak at 43' and 27" corresponds to the moment of detachment and is followed by a phase of low intensity, which should correspond to the initial stages of sliding.

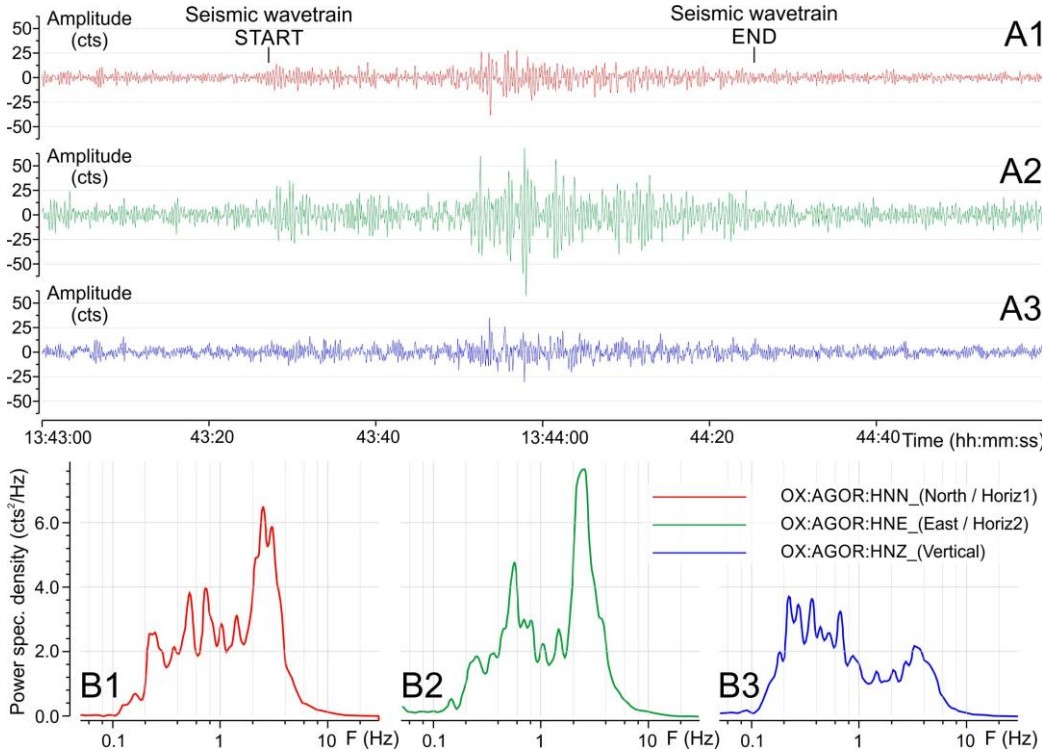

**Figure 15: Seismological records. (A)** Time series of the horizontal (A1 and A2) and vertical (A3) components of the ground motion induced by the Marmolada failure. **(B)** Power spectral density computed for the time series of the horizontal (B1 and B2) and vertical (B3) components.

The amplitude spectra of the different channels reveal the frequency signature of the wavetrain generated by the collapse (Fig. 15 B1–B3). The most energetic segment of the spectrum falls within the 0.2–4 Hz range. Most of the energy is concentrated in the horizontal components, as evidenced by the two highest-amplitude peaks (B1 and B2 in Fig. 15), both occurring at frequencies above 2 Hz. These peaks represent the impact of the ice mass on the underlying glacier after an

almost vertical fall of approximately 250 m. The horizontal components exhibit higher energy due to the inclination of the glacial surface in the impact zone, which exceeds 30 degrees. The spectral amplitude in the vertical component (B3 in Fig. 15) is lower in amplitude, confirming that most of the energy was released as transverse motion components. Similar observations can be made by analysing the spectrograms (time-frequency graphs) provided in the Supplementary Material.

### 4.5 Slope stability back analysis

The comprehensive characterization of the collapse zone led to the construction of a detailed pre-failure model of the glacieret (Fig. 16) that could be conceptualized for numerical modelling.

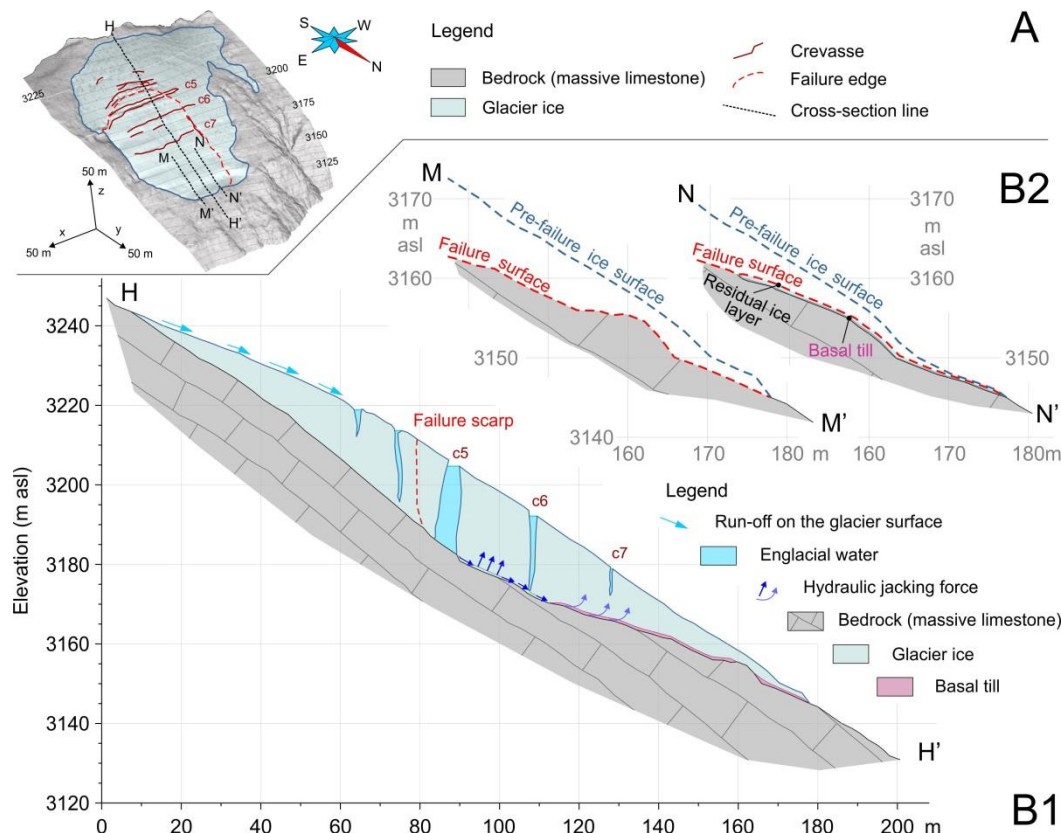

**Figure 16: (A)** Pre-failure model of the glacieret showing cross-sections HH', MM', and NN'. **(B1)** Pre-failure cross section HH' with indicated: failure surface; bedrock morphology; hydraulic jacking forces, etc. **(B2)** Post-failure models of the lower portion of the failure surface along cross sections MM' and NN'. The failure surface developed along an ice foliation in NN' while the detachment reached the underlying bedrock in MM'.

The long cross-section (cross-section HH' in Fig. 16B1) shows the actual failure scarp, along with the positions of the main crevasses and the thin basal moraine, which is mainly present in the medial and frontal sectors.

On the east side, the collapse involved the entire glacial body down to the bedrock, which was exposed after the failure (cross-section MM' in Fig. 16B2). The basal till, that was located at the base of the glacieret, was completely washed out. On the west side, the collapse occurred partly along a foliation plane and marginally at the bedrock interface (cross-section NN' in Fig. 16B2). The presence of the basal till was inferred from both RES images and direct observations of the residual ice surface on the western side of the post-failure niche (Bondesan and Francese, 2023). This basal layer appears to thin towards the west, with a variable thickness, generally estimated to be on the order of some decimetres.

The stability calculation was roughly performed along a cross-section that roughly follows the longitudinal axis of the glacier, partly shifted on the eastern side, where the rupture mostly occurred at the ice-rock interface.

**Table 1a.** Base scenarios (A, B or C) considered for numerical simulations.

| Scenario | | Type | HYP in crevasse | | | HJP | WL | | Factor of Safety (FoS) | | |
|---|---|---|---|---|---|---|---|---|---|---|---|
| | | | | | | | | | Computational method | | |
| | | | c5a | c5b | c6 | | Cohesion | Friction angle | Js | Jc | MP/GLE |
| | | | (kN/m$^2$) | | | (kN/m$^2$) | (kPa) | (degrees) | | | |
| A | 1 | HYP | 220 | | | | | | 4.25 | 4.58 | 4.38 |
| | 2 | HYP | 220 | 220 | | | | | 2.97 | 3.23 | 2.87 |
| | 3 | HYP | 220 | 220 | 200 | | | | 2.33 | 2.51 | 2.27 |
| B | | HJP | | | | 220 | | | 5.67 | 6.08 | 5.91 |
| C | 1 | WL | | | | | 208 | 21.6 | 5.84 | 6.23 | 5.86 |
| | 2 | WL | | | | | 156 | 19.2 | 5.16 | 5.50 | 5.40 |
| | 3 | WL | | | | | 104 | 16.8 | 4.42 | 4.70 | 5.27 |
| | 4 | WL | | | | | 52 | 14.4 | 3.97 | 4.22 | 4.86 |
| | 5 | WL | | | | | 10 | 13.0 | 3.56 | 3.75 | 3.66 |

A - HYP: hydrostatic pressure in crevasses (cx indicates crevasse id);
B - HJP: hydraulic jacking pressure;
C - WL: weak layer.

The short-term driving actions have been attributed to hydrostatic forces: (1) the hydrostatic pressure in the crevasses filled with water, (2) the hydraulic jacking pressure (Irvine-Fynn et al., 2011) and (3) the progressive reduction of friction along a thin weak layer located at the ice-bedrock interface. This weak layer could be referred to as a mixture of melt water, basal till, voids and ice (Zoet and Iverson, 2020; Huang et al., 2024).

The actual variability of the sliding surface was streamlined in the numerical model, and for the application of hydraulic jacking pressure and basal friction reduction, it was considered homogeneous from the base of the median crevasse to the glacieret's toe.

The driving action of these three triggering factors have been considered both individually (one at a time) and in combination (multiple factors acting simultaneously).

A first group of 9 numerical simulations (Table 1a) considered the separate contribution of each triggering factor (base scenarios), while a second group of 18 simulations (Table 1b) evaluated the combined effects of the various factors (combined scenarios). Three different computational methods were adopted for each simulation (see Supplementary Material for further details) resulting in 81 different slope stability models. In the base scenarios, the values of the Factor of Safety (FoS) were always greater than 1, corresponding to a stable condition, while in only three cases of the combined scenarios the FoS values were less than 1 (Table 1), evidencing the unstable condition.

**Table 1b.** Combined senarios (AiB and AiBCj) considered for numerical simulations.

| Scenario | | | Type | HYP in crevasse | | | HJP | WL | | Factor of Safety (FoS) | | |
|---|---|---|---|---|---|---|---|---|---|---|---|---|
| | | | | | | | | | | Computational method | | |
| | | | | c5a | c5b | c6 | | Cohesion | Friction angle | Js | Jc | MP/GLE |
| | | | | (kN/m$^2$) | | | (kN/m$^2$) | (kPa) | (degrees) | | | |
| AB | 1 | | A1+B | 220 | | | 220 | | | 3.25 | 3.49 | 3.25 |
| | 2 | | A2+B | 220 | 220 | | 220 | | | 2.31 | 2.48 | 2.32 |
| | 3 | | A3+B | 220 | 220 | 200 | 220 | | | 1.79 | 1.92 | 1.75 |
| ABC | 1 | 1 | A1+B+C1 | 220 | 220 | 200 | 220 | 208 | 21.6 | 3.19 | 3.42 | 3.25 |
| | | 2 | A1+B+C2 | 220 | 220 | 200 | 220 | 156 | 19.2 | 2.79 | 2.98 | 2.84 |
| | | 3 | A1+B+C3 | 220 | 220 | 200 | 220 | 104 | 16.8 | 2.40 | 2.56 | 2.54 |
| | | 4 | A1+B+C4 | 220 | 220 | 200 | 220 | 52 | 14.4 | 2.35 | 2.50 | 2.37 |
| | | 5 | A1+B+C5 | 220 | 220 | 200 | 220 | 10 | 13.0 | 2.20 | 2.34 | 2.28 |
| | 2 | 1 | A2+B+C1 | 220 | 220 | 200 | 220 | 208 | 21.6 | 2.24 | 2.41 | 2.27 |
| | | 2 | A2+B+C2 | 220 | 220 | 200 | 220 | 156 | 19.2 | 2.00 | 2.13 | 1.99 |
| | | 3 | A2+B+C3 | 220 | 220 | 200 | 220 | 104 | 16.8 | 1.62 | 1.72 | 1.69 |
| | | 4 | A2+B+C4 | 220 | 220 | 200 | 220 | 52 | 14.4 | 1.16 | 1.23 | 1.45 |
| | | 5 | A2+B+C5 | 220 | 220 | 200 | 220 | 10 | 13.0 | 0.71* | 0.75* | 1.19 |
| | 3 | 1 | A3+B+C1 | 220 | 220 | 200 | 220 | 208 | 21.6 | 1.71 | 1.84 | 1.75 |
| | | 2 | A3+B+C2 | 220 | 220 | 200 | 220 | 156 | 19.2 | 1.54 | 1.65 | 1.55 |
| | | 3 | A3+B+C3 | 220 | 220 | 200 | 220 | 104 | 16.8 | 1.24 | 1.32 | 1.30 |
| | | 4 | A3+B+C4 | 220 | 220 | 200 | 220 | 52 | 14.4 | 0.83* | 0.89* | 0.93* |
| | | 5 | A3+B+C5 | 220 | 220 | 200 | 220 | 10 | 13.0 | 0.42* | 0.44* | 0.71* |

A - HYP: hydrostatic pressure in crevasses (cx indicates crevasse id);
B - HJP: hydraulic jacking pressure;
C - WL: weak layer;
* Factor of Safety (FoS) < 1.

## 5. Discussion

The post-LIA retreat and disaggregation of the originally polythermal Marmolada glacier led to the detachment and transformation of part of its former firn area into a smaller glacier. As a result of ongoing global warming (Gilbert et al., 2010; Zemp et al., 2015), this smaller glacier further fragmented into minor units, one of which has recently become isolated from the main ice body, forming the glacieret that recently collapsed. It is important to emphasize that certain thermal effects can act as both predisposing and triggering factors. In particular, dynamic thermal processes—such as heat transported to the glacier base by meltwater percolating through crevasses, and heat absorbed by exposed rock surfaces—tend to function more as triggers than as underlying conditions.

### 5.1 Predisposing factors at the detachment site

The longer-term conditioning/predisposing factors are mostly determined by the relationships between the geometry of the glacieret and that of the carbonate bedrock. Further predisposing factors depend on the internal structure of the glacieret itself, on the physics and stratigraphy of the basal layers, the permeability of the frozen sub- and peri-glacial rocks, and on the amount of winter-spring snow precipitation, which affects the timing of glacier ice exposure during the summer.

### 5.1.1 Bedrock geometry and glacier structure

The glacieret occupied a small bedrock niche on the slope, partially suspended above a sub-vertical wall (Figs.11-13). The underlying bedrock surface is steep, with an average slope of 33° below the crest, 39° along the failure edge, and 20° in the middle and lower portions of the failure surface. The steep bedrock topography is often considered a key predisposing factor for the initial sliding (Faillettaz et al., 2011), especially when coupled with specific thermal conditions at the bedrock (Röthlisberger, 1981).

Just below Rocca Peak, the annual rate of glacial surface lowering was 0.58 m from 1990 to 2022. At the same time, increased downward flow caused crevasses to progressively open, partly dividing the ice body into blocks, increasing instability, and creating additional storage for surface meltwater. In particular, the large transversal crevasse, which extends down to the underlying bedrock, has effectively divided the eastern part of the glacieret into two blocks thus lowering the overall stability.

Many foliations are visible in the glacial body and they represent ductile structures that could play a major role in both water-routing (Jennings and Hambrey, 2021) and the development of a failure mechanism. Foliations, which could result from migration of air bubbles and fine debris (Hooke and Hudleston, 1978), are best depicted on the sub-vertical ice face of the fracture scarp, where their spatial extent—estimated through direct observation—reaches several decimeters. (see Supplementary Material). The high englacial RES reflectivity of the foliation planes indicates the presence of a significant 575 contrast in electrical impedance and therefore the presence of major sub-horizontal discontinuities in the ice body that affected the development of the failure plane during rupture. The thin basal layers interposed between the ice and the underlying bedrock, affect the overall stability of the glacieret. Unfortunately there is very little evidence about the existence and spatial distribution of this basal layer along the failure surface, with the exception of some deposits visible on the residual front of the glacieret (marked tbt in Fig. 9B).

### 580 5.1.2 Snow cover

The winter 2021-2022 was characterized by a total of new snow, from October to May, which was among the 10 lowest in the last 100 years (Figs. 5-7), as evidenced by the SAI anomaly index calculated on 8 stations in the Dolomites. Unfortunately, there are no snow thickness sensors at the PRC station, but ~30 years of data (1991-2020) from the nearby RVL station could be utilized as a reference and data from RVL (2615 m asl) and PZB (2905 m asl) stations, in the winter

2021-2022, could be used for comparison instead of PRC. The snow cover in RVL highlights the early winter snowfall and the following long period with little thickness of snow on the ground (about 0.5 m), which was increased by precipitation in mid-February. The snow cover increased again in late winter with the interposition of a melting period in the middle of April. This late-winter snow melted rapidly due to the mild temperature of the second third of May onwards. In general, fresh snow is highly reflective and therefore absorbs less solar radiation than ice, but on the other hand it is much less

compact and therefore melts more quickly. In the Marmolada case the unpacked snow was prone to accelerated melting with high production of liquid water as compared to the typical compacted winter snow. In addition, due to the high porosity of the recent snow layer (Clifton et al., 2007), there was very little surface runoff and melting water penetrate into the glacier body as inferred by the satellite images taken two weeks before the failure. The snow cover in PZB has a similar trend to RVL despite this station (facing the Marmolada glacier) being located ~300 m higher in elevation than RVL.

**5.1.3 Permafrost and ice thermal conditions**

Evolution of thermal conditions around the failure zone is crucial to get better insight into potential collapse mechanisms but involves complex transient glacier-permafrost interactions (Figs. 7-8). Where bedrock of the northern slope has remained free of glaciers during past decades, its temperature can be estimated at a few degrees Celsius below freezing, ranging from typically near 0°C at lower altitudes, around 2400-2600 m asl, and reaching -1°C to -4°C towards the uppermost parts. With

such surface temperatures, permafrost depth may in places exceed several tens of m (Etzelmüller et al., 2020). In the Alps, permeable firn zones, warmed up by percolating and refreezing meltwater, are temperate up to altitudes between approximately 3400 and 3900 m above sea level (Haeberl\i and Alean 1985; Suter et al., 2001; Bohleber, 2019) while the impermeable ice of the ablation zones in areas with permafrost tends to be cold. The LIA glacier at the Marmolada site with extensive warm firn areas may therefore have been polythermal and predominantly warm-based (cf., Wilson and Flowers,

2013) with only its lowest margins containing cold ice, partially frozen to bedrock. Higher parts of bedrock underneath the LIA glacier could therefore have remained largely unfrozen. Interestingly, ice temperatures recorded several tens of cm within the walls of some WWI tunnels, dug in the Marmolada glacier in 1917 (Hess, 1940), in the elevation range 2800-3200 m asl, already showed average values of -1.32°C in tunnel "32" (located in the vicinity of the failure site) and of -1.27°C in tunnel "S" (located ~700 m west of the failure site). These values suggest that moderately cold conditions prevailed at that

time on the northern side of the Marmolada. Over the following decades, the progressive loss of insulating warm firn areas likely led to further cooling of surface ice. This cooling, in turn, is thought to have affected the thermal regime of the ice body from which the avalanche later detached.

The long-term thermal evolution of the glacieret has, on one hand, caused the freezing of its ice base to the underlying rock, as confirmed by temperature measurements taken in the ice borehole throughout the upper portion last summer. On the other

hand, the warming-induced permafrost degradation (Gruber and Haeberli, 2007), suggested by the fifteen-year temperature and active layer and permafrost thickness monitoring in the rock borehole at PZB (Fig. 7), is likely a contributing factor to weakening the basal interface in the frontal zone of the glacier. The curved geometry of the median crevasse, clearly visible

after the failure, suggests that the glacier was primarily deformed under the influence of gravitational forces, with negligible basal sliding. This further supports the hypothesis of a locally cold ice base. Moreover, as clearly shown in Fig. 9B, a large portion of basal ice remained frozen to bedrock after the failure (see more details in subsection 5.2.2). Such recent subglacial freezing may likely have influenced the hydraulic permeability of the karstic subglacial rocks, potentially reducing drainage capacity beneath the cold ice. This permeability reduction could have helped building up high water pressures in the ice body.

## 5.2 Triggering factors

The trigger factors are closely interrelated with the conditioning factors, as the trigger factors considered individually would not have been sufficient to induce the collapse.

### 5.2.1. Seismic shock

Among the possible triggers, we also evaluated whether a small earthquake could have caused the displacement of an ice mass that was already near its stability threshold, thereby leading to the full failure (Figs. 14-15). Such occurrences are quite frequent (Podolskiy et al., 2010) and the hypothesis was likely suggested by analogy to events associated with the recent Gorkha earthquake (Pettenati et al., 2023) that affected the Kathmandu valley in the Himalaya region in April 2015, causing the Langtang Valley disaster (Fujita et al., 2017). The same earthquake also triggered an ice avalanche (Kargel et al., 2016) that struck Everest base camp killing 24 people and injuring 61 causing the deadliest disaster on the mountain ever. However, on July 3, 2022, the NOAN seismic network reports only three events which are far away from the Marmolada glacier (Table 2). Two events occurred several hours before, while the third event occurred approximately two hours later. No suspicious seismic activity was recorded nearby the Marmolada glacier on the day of the collapse.

**Table 2.** Seismic events occurred on July 3, 2022 in northern Italy and surrounding regions.

| Site | Region | Magnitude $M_L$ | CEST hh:mm:ss | Latitude decimal degrees N | Longitude decimal degrees E |
|---|---|---|---|---|---|
| Forni di Sotto (F. Venezia-Giulia) | Eastern Alps | 1.0 | 6:33:40 | 46.336 | 12.621 |
| Giogo dello Stelvio (Alto-Adige) | Central Alps | 0.5 | 7:18:44 | 46.558 | 10.482 |
| Sassuolo (Emilia-Romagna) | N. Apennines | 2.3 | 17:53:25 | 44.545 | 10.798 |

The seismological localization of the Marmolada collapse and avalanche is accurate, as it corresponds closely to the actual position within a reasonable error range, despite the absence of seismic stations in the northeast quadrant. The recorded waveforms are dominated by surface waves, indicating that the rupture occurred very near the surface, and they show the typical response associated with an avalanche or an ice-rock fall.

The matching filter method (Gibbons and Ringdal, 2006; Sugan et al., 2014) was applied to the time series from the AGOR station to further dispel any uncertainties that an earthquake triggered the collapse. The collapse-related time series

considered the two minutes of the event itself and the three preceding minutes. Two low-magnitude seismic shocks occurred in 2023 (M0.9 on April 19[th] at 22:12:50 and M1.3 on July 10[th] at 22:47:41) were used as the template for the cross-correlation. The matching process resulted in a degree of cross-correlation that did not exceed the value of 0.40, confirming that the failure was not triggered by a seismic shock.

Neglecting the seismic shock, the overall triggering factor should be sought in a complex interaction between the extremely high air temperatures, which lasted for more than two months, and the resulting thermo-hydraulic conditions of the ice of the glacieret and of the bedrock.

### 5.2.2 Thermal conditions

There is no doubt about the temperature anomaly (Figs. 5-6) that characterized the late spring and early summer of 2022. At
PRC station the 70 days preceding the collapse showed monthly temperatures (max, avg and min) not only above the standard deviation range for the 30-year period 1990-2020 but these values turned out to be absolute maxima. The temperature in the PZB borehole on July 3, 2022, reaches almost an absolute maximum near the surface, while it is an absolute maximum in the depth range between 0.5 m and 3.5 m.

The effect of elevated temperatures is twofold: it caused an intense heatwave over the glacier (Chen et al., 2023) and
impacted the exposed bedrock, affecting both the production of meltwater and the progressive warming of the bedrock. A rough estimate (Bondesan and Francese, 2023) indicates that around 15,000 m$^3$ of melt water had infiltrated the glacier since mid-May. This prolonged heating must be considered in relation to longer-term conditioning factors, such as ice temperature and sub-glacial permafrost.

The meltwater filled the crevasses and warmed the ice down to their bottom, particularly at the large median crevasse that
reaches the underlying bedrock. The melting may also have been also accelerated by the strong north-south thermal contrast between the warm south face and the cold northern slope of the mountain that caused an heat flow across the mountain ridge (Noetzli et al., 2007).

A detailed analysis of the failure surface provides further information on the thermal conditions of the coupled bedrock-ice system at the time of the failure (Figs. 9-10). The failure occurred along a rather complex surface.
Considering the frontal wedge, it is evident that, to the west, at least one-third of the failure plane developed along an envelope surface of foliation planes (area marked a0 in Fig. 9B), with the remaining portion along a vertical discontinuity on the western flank. To the east the detachment is right at the ice-bedrock interface (area marked e0 in Fig. 9B) with the bedrock clearly exposed along the entire western shoulder. To the south a thin residual layer of basal ice, due to meltwater refreezing at the bottom of the crevasse, was left on top of bedrock after the failure (area marked s0 in Fig. 9B). It took
several days before this thin layer melted completely, as documented in several videos taken in the weeks after the event. This rear portion of the sliding surface is precisely the area surrounding the foot of the crevasse where the ice in the basal layer underwent phase changes, eventually leading to conditions of instability.

Similar to the failure of the Altels Glacier in 1895 (Faillettaz et al., 2011), a large portion of basal ice is clearly frozen to the bedrock, at least at the northwest front (labeled 'a0' in Fig. 9B).

The residual portion of the glacieret is comprised of moderately cold ice, consistent with the prediction curve (Haeberli et al., 2004a) and with the borehole measurements taken during the summer of 2024.

### 5.2.3 Englacial drainage network

Regarding the presence of flowing melt water at the base of the glacier, some observations could be made. There is no evidence of a frontal discharge in the aerial image (SAT, 2022) and in the satellite images (Figs 9-10) taken the day before
and a few days before the collapse respectively. This indicates that the meltwater is not routed in a channelized subglacial drainage, as it is common for larger warm-based glaciers (Röthlisberger and Lang, 1987; Richards et al., 1996). On the contrary, there is clear evidence of water outflowing from the eastern edge of the large median crevasse routed along a bédière (marked with letter b in Fig. 10A), confirming that an englacial drainage network is not developed. Englacial water is present in the glacieret but its cold ice frozen to permafrost rocks prevents basal or subglacial water flow (Paterson, 1994).

### 690 5.3 Numerical simulations

Insights into the significance of the driving forces that triggered the failure can also be obtained from numerical simulations of glacier slope stability (Figs 16-18).

The ice shear strength parameters (cohesion and friction angle) were inferred from the available literature. Seaki et al. (1985) conducted a multi-temporal analysis on cohesion and friction angle of sea ice over a six-year period and in different
temperature conditions. Tang et al. (2024) carried out direct shear tests on polycrystalline ice at temperatures ranging between -0.5 and -40 °C, obtaining Mohr-Coulomb shear strength parameters. Available temperature data suggest that the upper portion of the Marmolada Glacier, at least above 3000 meters in elevation, is polythermal (Bondesan and Francese, 2023; Boeckli et al., 2012). Polythermal ice conditions were encountered during the summer campaigns in the Ortler-Cevedale massif, approximately at the same elevation (Francese et al., 2019). Based on the above considerations and known
ice temperatures of Alpine sites (Fig. 8B), an ice temperature ranging from -2.5°C to -4°C was selected to estimate the appropriate parameters and carry out the numerical simulations. These initial values were confirmed by borehole temperature measurements taken from the ice at the failure site in August 2024 (Fig. 12C).

Equivalent parameters for the Marmolada limestone were also inferred from the available literature and particularly from a systematic classification of the Dolomitic rocks (Longo, 2018) based on the RMR - Rock Mass Rating System (Bieniawski,
705 1989).

The overall failure mechanism can be reasonably interpreted as a simple frictional model of a solid mass sliding along a surface, and for this reason the LEM was used. The 2D approach used for the simulation has certainly some limitation but the currently available parameters prevent a meaningful 3D approach. In addition, it is worth mentioning that the failed ice body, to the west, was firmly joined to the western glacial shoulder, which did not collapse. It is still possible that the rupture

along the sub-vertical portion of the western surface was influenced by the presence of a longitudinal discontinuity. In some of the aerial and satellite images from previous years, the presence of a longitudinal fracture was indeed visible right at the location of this rupture. This limitation could results in an underestimation of the FoS values calculated via a simple 2D approach.

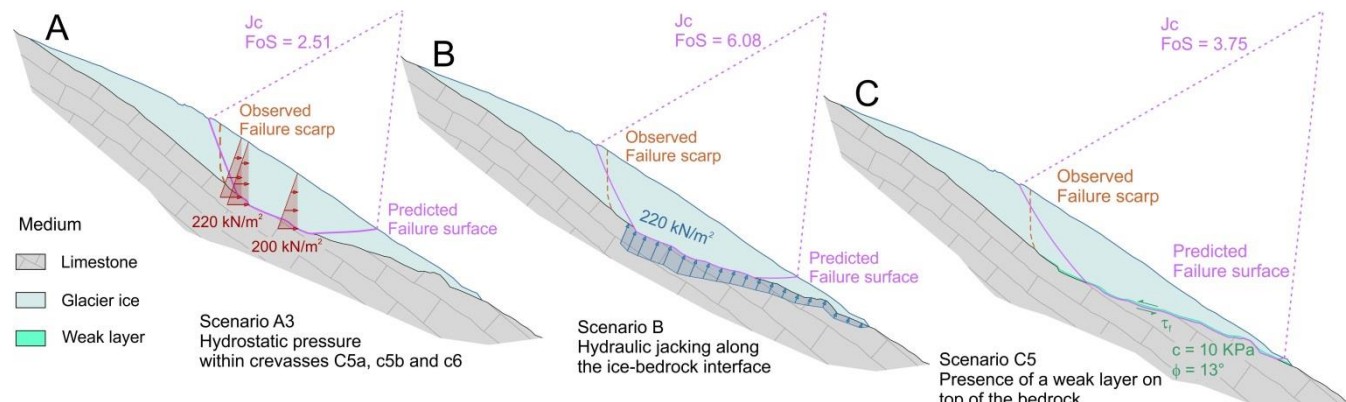

**Figure 17: Numerical simulations: base scenarios. (A) Scenario A3 - hydrostatic pressure acting on crevasses 5a, 5b and 6; (B) Scenario B - hydraulic jacking pressure acting along the ice-bedrock interface; (C) Scenario C - reduction of the basal friction caused by a weak layer on top of the bedrock in the worst case (c=10 kPa; $\phi$=13°). See text and Table 1a for details.**

The base scenarios, conceptualized in Fig. 17, weights the effects of three single causes of instability: hydrostatic pressure in
crevasses, hydraulic jacking pressure, and presence of a weak layer at the base. The nature of this weak layer is still under investigation, but available data provide several indications. The presence of a water layer can be ruled out, as aerial and satellite images indicate no frontal discharge, excluding the presence of a developed basal drainage network. Among the possible alternatives, the most probable is a softening of a thin basal layer, caused by pressurized water (Church et al., 2021) stored in the median crevasse, forcing its way through it and forming a locally connected drainage system (Kavanaugh and
Clarke, 2001). While other alternatives are possible, this scenario is the most likely. The process led to the formation of a multi-decimetric basal layer composed of a mixture of water, ice, debris, and voids, resulting in a reduction of shear strength, which allowed shear stress to easily exceed the resisting forces, thereby causing instability. Moreover, warm permafrost with temperatures between about 0°C and -1.5°C contains ice with unfrozen water. Such "warm permafrost" is weaker not only than colder but also than unfrozen bedrock (Davies et al., 2001).

In the three base scenarios (Table 1a and Fig. 17) the minimum FoS is significantly above the value of 1 for the three different computational methods. The values of FoS computed with the different methods are more or less comparable. The "Janbu corrected" (Jc) will therefore be considered for the purpose of this discussion.

The summed effects of the hydrostatic pressure in multiple crevasses lower the FoS to a minimum value of 2.51 (Fig. 17A) while the effect of the hydraulic jacking pressure alone seems to be negligible, as the FoS value is significantly higher than
the previous scenarios indicating overall stability (Fig. 17B). The weak layer alone does not generate instability also when

modelled using the minimal values for the c-ϕ (cohesion - friction angle) pairs (Fig. 17C) as it results in a FoS of 3.75. The predicted failure surface is surprisingly close to reality in the case of hydraulic jacking pressure alone (Fig. 17C and Fig. 7B), albeit along a cross section shifted less than 10 m westward compared to the cross-section modelled in the simulations.

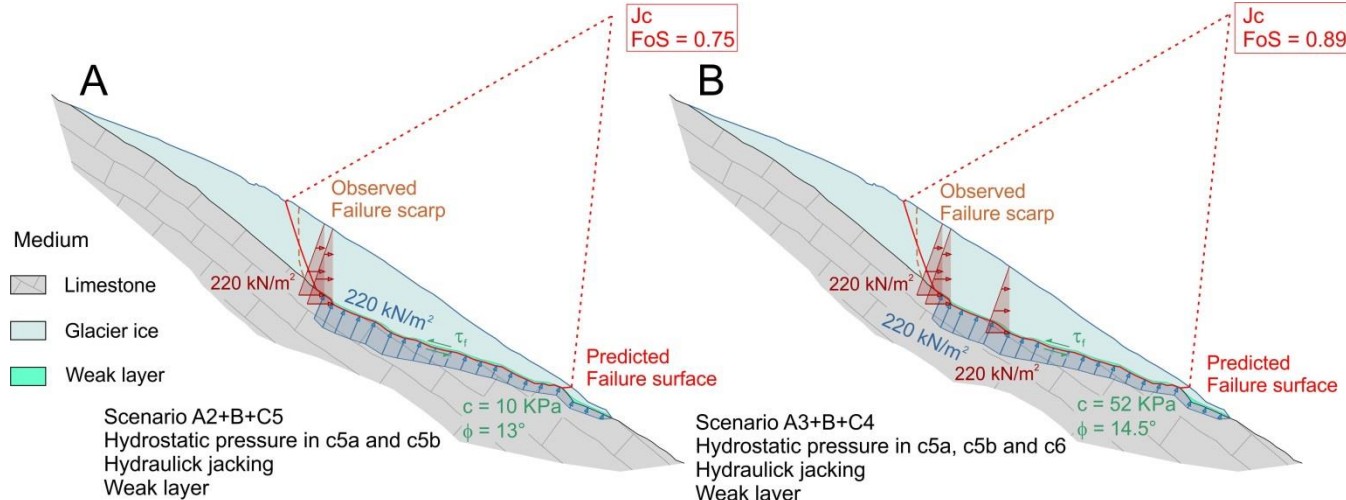

**Figure 18: Numerical simulations: combined scenarios. (A) The instability condition, with relative FoS less than 1, is reached in the scenario A2 (hydrostatic pressure in two crevasses) + B (hydraulic jacking pressure) + C5 (worst c, ϕ pair in the weak layer); (B) The instability condition, with relative FoS less than 1, is reached in the scenario A3 (hydrostatic pressure in all crevasses) + B (hydraulic jacking pressure) + C4 (second worst c, ϕ pair in the weak layer). See text and Table 1b for details.**

The minimum FoS (2.51) in the three basic scenarios corresponds to the hydrostatic pressure condition acting in all crevasses. Therefore, it seems possible to exclude the single driving action for the triggering of the collapse and this finding is consistent with what happened in other similar cases (Mortara and Dutto, 1990; Haeberli et al., 2004b; Faillettaz et al., 2015). The FoS was always significantly greater than 1 in the 27 simulations in the group of the base scenarios.

The combined scenarios weight the effect of different associations of the factors of instability. The FoS is always greater
than 1 in case of hydrostatic pressure associated with hydraulic jacking pressure (scenarios Ai+B in Table 1b) with a minimum of 1.92, suggesting that the driving action of meltwater alone cannot induce the failure. The inclusion of the weak layer determines conditions of instability (in combination at least with scenario A2, which considers the hydrostatic pressure in two of the three crevasses) but only in the condition of minimum resistance (scenario C5) of the weak layer itself (Fig. 18A). The FoS in the A2+B+C5 scenario drops to 0.75. The predicted failure surface almost reaches the front of the
glacieret. This result is consistent with the model used for the simulations, in which the weak layer was extended down to the front of the glacieret. This predicted failure surface corresponds to reality for the ice volume located in the eastern half of the glacieret (Fig. 18B). Considering finally the hydrostatic pressure acting on all three crevasses (scenario A3+B+Cj in Table 1b), instability already occurs starting from a weak layer characterized by a couple of parameters c-ϕ (C4) which is the

second most unfavourable among those modelled (Fig. 18B). The FoS in the A2+B+C5 scenario drops to 0.89. The FoS was less than 1 in just 8 of the 54 simulations in the group of the combined scenarios.

The considered triggering causes are almost equivalent as none alone prevails over the others. However, it could be possible to establish a "ranking", considering the values of FoS, based on how close they are to one, although they are greater than one. In this sense, the worst triggering factor relates to the water-filled crevasses (FoS = 2.51, lower than the others). It is followed by the weak layer characterized by the worst parameters (FoS = 3.75). The least important is hydraulic jacking (FoS = 6.08). However, there are many uncertainties about the parameters of the weak layer.

As for the predisposing factors, the initial formation of deep crevasses in cold ice frozen to permafrost rocks appears to be the most critical condition, as water that seasonally penetrates the crevasse can lead to its progressive deepening, transferring heat to the deeper layers of the glacier (Gilbert et al., 2020). Without their presence, hydraulic jacking would be more difficult to initiate and, as a chain, the degradation of the basal (weak) layer would also be slower. Moreover, the loss of shear strength of the basal interface is connected with the decrease in effective stresses due to the increase in water pressure. In other words, it can be assumed that the other predisposing/triggering factors also depend upon the development of the crevasses.

The "sine qua non" condition for the instability of the ice mass that collapsed on 3 July 2022 is then a combined action of three instability factors, none of which alone determines a FoS condition lower than 1.

Particularly anomalous temperature conditions also occurred during the spring and summer of 2003. The only difference was a shorter duration and values falling outside the standard deviation range for the reference period 1990–2020 only in the month of June, when the monthly maximum temperature was 3.5°C higher than the average for the reference period (in June 2022 the positive difference was 4.3 °C). On that occasion as well, as evidenced by satellite images, the large median crevasse was completely filled with meltwater, which overflowed from the eastern edge and generated a bédière (Bondesan and Francese, 2023). In 2003, instability did not occur, most likely due to the smaller volume of meltwater, which—by penetrating the crevasse—warmed the basal layer, and due to the greater extent of the glacieret, which, despite the thicker ice, was well frozen to its bed throughout the entire frontal area.

Finally, some considerations must be made on the predictability of the glacial collapses. It is known that cold-based glaciers show very few precursors detectable at the surface (Röthlisberger, 1981), while in case of warm-based glaciers some surface effects could be observed in the imminence of the break-off (Faillettaz et al., 2011). Surface velocity is the parameter to be monitored but, in general, the time window for predicting a failure is very short and no longer than a few days or at best weeks (Pralong et al., 2005), with the danger of false alarms. Quantitative glacier monitoring is probably more effective than visual glacier observations but it is also more costly and then possibility of false alarms must be taken into account. Glacier seismicity (Walter et al., 2009) could be used to correlate ice quakes with ice dynamics but this seismicity is also controlled by surface melting and changes in subglacial water pressure (Mikesell et al., 2012), resulting in inherent difficulty in transposing ice quake occurrences into an effective collapse precursor.

An alternative strategy may be to create hazard scenarios based on sets of key parameters (e.g. temperature, snow cover, melting rate, etc.) to be cost-effectively collected or reliably estimated but only after having achieved a full knowledge on glacier (including major crevasses) and bedrock morphology. A good control on the glacier geometry allows for the use of

advanced ice flow models (Zekollari et al., 2022) that could be turned into effective numerical simulations of stability. Current technology allows the simple and cost-effective acquisition of time-lapse geophysical images of almost all Alpine glaciers (Ruols et al., 2023) via UAV-mounted broadband RES antennas, which could be flown just above the glacier surface, preventing the loss of resolution that generally affects helicopter-based RES surveys.

## 5. Conclusions

The failure of the Marmolada glacier resulted from an unfortunate combination of conditioning/predisposing and triggering factors, which, acting over different time frames and with varying weights, favoured instability under the unusual atmospheric conditions that occurred in the Alps during the late spring and summer of 2022, ultimately leading to the collapse. A thorough compilation of information was used to define pre-failure settings providing the basis for numerical back-analysis.

The broken ice body, as a consequence of the warming-induced disaggregation of the Little Ice Age glacier became an isolated cold glacieret—consisting of massive, impermeable, yet crevassed ice—at least over the past three decades. The progressive opening of a large median crevasse increased the glacieret capability of storing englacial water. Its average ice temperature can be estimated at some -2.4°C, i.e., relatively close to melting conditions but with at least partially freezing or frozen bed. Additional conditioning factors included the steep slope inclination; the presence of low-angle discontinuities,

such as ice foliations and/or discontinuous basal till layers; and the complex thermal conditions at the ice–bedrock interface, which may be associated with permafrost degradation.

The most likely triggering factors are associated with minimal winter snowfall and a prolonged positive thermal anomaly. The limited thickness of low-permeability snow layers—combined with extreme air temperatures—allowed excess meltwater to penetrate into the glacier. This resulted in water filling deep crevasses and generating subglacial water pressures

exceeding floating conditions. The absence of a connected drainage network, because of basal ice frozen to bedrock at the glacier bottom, created the condition for the development of an increasing hydraulic over-pressure. Meltwater also contributed to heat exchange at the base of the large median crevasse, promoting—through refreezing—the softening process of the basal ice, which eventually resulted in water forcing its way beneath the glacieret and, ultimately, in reducing the shear strength within its thin complex basal layer comprised of ice, water, debris and voids.

In short, the base of the cold ice in the failure area was most probably frozen, presumably for many decades already. This frozen condition largely, if not entirely, prevented subglacial water drainage. The recent opening of a full-depth crevasse allowed water to get to the base of the ice and to over-pressure the system, progressively forming a gap that led to hydraulic jacking and a loss of basal friction. Continued atmospheric warming leading to a general degradation of permafrost in the

area may have led to a warming of the ice base near the frontal and marginal area of the glacieret. This plausible contributing

factor might help explain why the crevasse filling event in 2003 did not lead to a failure.

An earthquake, as the final triggering mechanism, can be excluded. Results from numerical simulations suggest that the triggering of the final collapse was most likely due to the simultaneous interaction of hydrostatic pressure, hydraulic jacking pressure, and a reduction in basal friction caused by the presence of a weak basal layer. This thin layer appears to be a key factor in the failure, as instability does not occur by hydraulic pressure alone.

Ranking of the predisposing/triggering factors accurately is not a straightforward task, but it's possible that the most influential predisposing factor is related to the progressive widening of crevasses while the crucial triggering factor could be associated with the water infill of crevasses along with the weakening of the basal layer.

Finally, it should be noted that some resisting forces, due to the real three-dimensional shape (i.e. lateral failure), were neglected, partly limiting the effectiveness of numerical simulations. The 3D morphology of the niche and the existence of

mild counter slopes in the calcareous bedrock are further aspects that could not be fully considered in this simple 2D conceptualization.

Future improvement of the numerical simulations will include a full 3D modelling of the failure, based on new borehole measurements of the ice and bedrock temperatures and high-resolution RES geophysical images data that are planned to be collected.

**CRediT authorship contribution statement**

RF and AB both designed the study, organized, and participated in geophysical field surveys along with SP and MG. RF re-organized, and re-structured the various data types implementing the GIS and the database. GR processed the satellite images. WH and MV provided information on climatic variables and permafrost. FP and DR processed the seismological data. RV implemented the numerical models and conducted the simulations. All the authors equally contributed to the

analysis of the results and to the writing of the manuscript.

**Declaration of competing interest**

The authors declare that they have no known competing financial interests or personal relationships that could have appeared to influence the work reported in this paper.

**Funding**

Research was funded by: FFABR 2017 (MIUR, Francese); FIL 2019 (University of Parma, Francese); DOR (Dotazione Ordinaria Ricerca) 2015–2021 (University of Padova, Bondesan); PRAT2013 (Progetto di Ateneo 2013, University of Padova, Bondesan). Other funds were provided by the National Institute of Oceanography and Applied Geophysics (OGS).

**Data availability**

Topographical, RES, seismological, IR, and borehole temperature data are available from the authors. The meteorological
time series (temperature, snow cover, and rainfall) are publicly available, but a request must be submitted to the Italian agency ARPAV (https://www.arpa.veneto.it/).

**Acknowledgments**

The authors gratefully thank Nicola Casagli (UNIFI) for providing and processing TLS data collected just after the collapse. The authors are also grateful to Riccardo Percacci for his help and suggestions in applying the matched-filter method.
Particular thanks to Nuccio Bucceri of Land & Technology Srl for his aid in processing the IR images. Special thanks to Provincia di Trento and to the cartographic bureau of Regione del Veneto for providing past satellite/aerial imagery and LIDAR data and to Alberto Carton for providing historical maps. Additional thanks to the Provincia di Trento for providing the steam probe for ice drilling. AB is a research fellow at the University of Stellenbosch, and RF is a research fellow at the National Institute of Oceanography and Applied Geophysics. We thank Dr. Martin Truffer and the anonymous reviewer for
their constructive comments, which significantly improved the quality of this manuscript.

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
