# Peer review of "Failure of Marmolada Glacier (Dolomites, Italy) in 2022: Data-based back analysis of possible collapse mechanisms"

_Natural Hazards and Earth System Sciences, 2024_

## Author Comment (AC1)

**Reply to reviewer #1 (Dr. Martin Truffer)**

This paper has a lot of interesting information about the catastrophic glacier failure of a piece of the Marmolada Glacier that caused several fatalities and must be rated as one of the bigger glacier related natural disasters in Europe. With the rapid change of glaciers in the current climate, it is important to better understand the mechanisms that lead to such an event. The paper does a good job in assembling a variety of data sets that allow for some conclusions. However, the paper would benefit from a significant reorganization and more detailed analysis. Parts of the paper go into great detail about things that may only be peripheral (parts of the permafrost discussion), while being really sparse on things that matter significantly (e.g. how the LEM model works). This review is rather long, because I think the paper needs major work. However, a clear paper on this topic is an important contribution to the literature and I hope it can be put into publishable form.

*We thank Martin Truffer for his extensive and constructive feedback.*

*The paper has been partially reorganized. The permafrost analysis has been simplified, while greater emphasis has been placed on the description of the LEM model. The authors acknowledge that several hypotheses were proposed without direct proof; however, this is an inherent part of the process of understanding natural phenomena. Moreover, these hypotheses have the potential to stimulate further discussion. Nevertheless, as per the reviewer's suggestion, the majority of the speculative statements have been removed, particularly in cases where the available data do not allow for reasonable estimate of the weight of the different controlling variables on the failure.*

*Regarding the different sections: the 'Introduction' has been slightly shortened to better focus on the topic; the 'General Settings' section has been expanded to provide a more detailed geological context of the site; the 'Results' section has been condensed, while the 'Discussion' has been expanded to better constrain the predisposing and triggering variables. The 'Abstract' was also modified avoiding speculations and now it reads:*

*P0*

*"A small, isolated portion of the Marmolada glacier broke off on July 3, 2022. The detached ice mass had an estimated volume of 70,400 m3 and slid down the slope killing 11 mountaineers after having travelled for approximately 2.3 km along the northern slope. This event is considered among the deadliest ice avalanches historically recorded in the Alps.*

*The unusually high air temperatures in late spring and early summer of that year led to an excess of meltwater, which, since mid-June, overpressurized the englacial discharge network, partly blocked due to frozen conditions at its base. Cold ice and sub-glacial permafrost were among the primary variables controlling the thermal state of the sliding plane.*

*The cause of the collapse was investigated implementing a conceptual model that was further corroborated through simplified numerical simulations using the Limit Equilibrium Method. Pre- and post-failure satellite and aerial images, laser mapping, geophysics and morpho-climatic data were gathered in a comprehensive database and analysed to better understand the role and interaction of the predisposing and triggering factors as well as their mutual intercation. Particular attention was given to reconstructing the varying conditions of the failure surface, which partly developed along ice foliations near the glacieret's base and partly right at the ice-bedrock interface. An earthquake triggering the failure was excluded based on the processing of the available seismological observations.*

*It resulted that none of the three forces considered in the numerical analysis—namely, hydrostatic pressure in crevasses, hydraulic jacking, and basal friction reduction—individually caused the condition of instability. To reach this condition, it was necessary to invoke a combination of these actions, for which it was finally possible to estimate their relative weights"*

*A paragraph was added at the end of the Introduction to clarify the core framework of the paper (see P1 below).*

*All the figures have been updated based on the reviewers' comments. The 3D model in Figure 2 has been replaced with a map view (as suggested by reviewer #2). Subpanel C has been removed from Figure 8. Figure 11 (GPR scans) has been divided into two separate figures and a migrated version of the bedrock response was included. The time series from the Everest avalanche have been removed from Figure 14 and replaced with the spectral components of the Marmolada failure. Figure 16 has been removed. The captions have been revised and strengthened. The font size was increased in all figures to improve readability.*

The permafrost discussion is confusing. It appears that you argue that this glacier used to be temperate with a firn layer. Then it lost its firn layer and became cold-based. Then it started warming at the base again and weaken the ice, partially due to warming air temperatures and also heat infiltration through bed rock from the south facing side. While all of these are plausible at some level, this could be presented much more consistently. The Boeckli et al map is useful for context, but ultimately, the map and the nearby borehole will not allow much of an assessment of the thermal state. Neither do the thermal surface measurements. Those were acquired at a time when it could simply be cold surface temperatures that say nothing about the thermal state of the ground a few meters down. The most useful measurement is the borehole in the ice. This needs to be emphasized more. It clearly shows a cold ice base, although the base is barely below the level of where seasonal fluctuations would affect temperature.

*The permafrost aspect is indeed complex, involving a complicated transient thermal interaction between surface and subsurface conditions. We try to make it as clear as possible. We reworked the introductory sentence writing:*

*P1*

*"We used a data-based back-analysis approach to infer the basal properties of the failure surface, aiming to understand the critical interactions among englacial water (which altered temperature and pressure fields within the glacier and at its base), permafrost in rocks and sediments (Noetzli and Gruber, 2009; Rossi et al., 2022), the glacier's thermal state, and most importantly the existence of a thin, heterogeneous, and discontinuous layer at the ice-bedrock interface (Zoet and Iverson, 2020; Huang et al., 2024)."*

*And later in the Materials & Methods:*

*P2*

*"The permafrost information for the study area was inferred from the 25 m x 25 m model for the Alpine chain (Boeckli et al., 2012) and from the nearby borehole of the Piz Boè (PZB) equipped with operational temperature sensors since the year 2011 (Crepaz et al., 2011).*

*The model devised by Boeckli (2012) is based on Mean Annual Air Temperature (MAAT) of the period 1960-1990. PZB data confirm this model and provide some information about ongoing permafrost warming in the region. The PZB borehole was drilled in 2011 at 2905 m asl, which is about 300 m lower than the failure site, on the warm eastern slope of the Piz Boè massif, about 8 km to the NNW.*

*Ice temperature estimates for the Marmolada glacier were inferred by using and updating published data (Haeberli et al., 2004a; Fischer et al., 2022). In early August 2024, a borehole was drilled in the residual ice body just above the failure scarp using a steam-based drill bit and was equipped with four temperature sensors. The sensors were located just above the ice-bedrock interface (at a depth of 11.5 m below the surface) and at -2.5 m, -5.5 m, -8.5 m from the ice surface respectively (Fig. 12)."*

*And in the Discussion:*

*P3*

*"Evolution of thermal conditions around the failure zone is crucial to get better insight in possible collapse mechanisms but involves complex transient glacier-permafrost interactions. Where bedrock of the northern slope has remained free of glaciers during past decades, its temperature can be estimated at a few °C below freezing temperature, probably close to 0°C at lower altitudes, around 2400-2600 m asl, and reaching -1°C to -4°C towards the uppermost parts. With such surface temperatures, permafrost depth may in places exceed several tens of m (Etzelmüller et al., 2020; Rossi et al., 2022).*

*In the Alps, pervious firn zones, warmed up by percolating and refreezing meltwater, are temperate up to altitudes between approximately 3400 and 3900 m above sea level (Haeberli and Alean 1985; Suter et al., 2001; Bohleber, 2019) while the impermeable ice of the ablation zones in areas with permafrost tends to be cold. The LIA glacier at the Marmolada site (Fig. 8A) with extensive warm firn areas may therefore have been polythermal and predominantly warm-based (cf., Wilson and Flowers, 2013) with only its lowest margins containing cold ice, partially frozen to bedrock. Higher parts of bedrock underneath the LIA glacier could therefore have been largely unfrozen. Interestingly, ice temperatures recorded several*

*tens of cm within the walls of some WWI tunnels, dug in the Marmolada glacier in 1917 (Hess, 1940), in the elevation range 2800-3200 m asl, showed average values of -1.32°C in tunnel "32" (located in the vicinity of the failure site) and of -1.27°C in tunnel "S" (located ~700 m west of the failure site), i.e., moderately cold conditions on the northern side of the Marmolada. With the progressive loss of warm firn areas, such cooling of surface ice especially also affected the small ice body, from which the ice avalanche later detached. Such cooling of surface ice over time must have induced the formation of subglacial permafrost, where it had been absent under LIA conditions. Such recent subglacial freezing may likely have influenced the hydraulic permeability of the karstic subglacial rocks and, hence, the drainage mechanisms of the cold surface ice body."*

The influence of the southern slope on the other side of the mountain is an interesting hypothesis, but it is very difficult to see how that heat flux would have allowed the glacier base to first freeze and then later thaw again. There is a lot of potential for some simple thermal model here, but even in the absence of that, the various influencing factors and hypotheses need to be stated much more clearly.

*We acknowledge the reviewer's comments. Such strong asymmetries are quite characteristic near mountain ridges. The primary influence is not via heat conduction but could in many cases be through water infiltration. We nevertheless eliminated this aspect in order to focus on the already quite complex primary aspects (see P3 above). This paper does not include thermal modeling, as it was not our primary focus; however, it could be conducted in future works to gain a better understanding of the collapse.*

The model discussion also needs to be made clearer and many questions remain. First, little detail is given about the LEM model. Second, it is not clear to me whether the parameters from Huang et al are applicable here. That paper finds peak strength in shear tests that are conducted under very high constant strain rates. Applying such high strain rates leads to stresses that are very high and to a quick ductile/brittle transformation. However, in the situation of this glacier, the geometry imposes certain stresses that would lead to deformation of ice, and, in this case, failure. The high stresses obtained in the Huang paper probably explain why it is so difficult to obtain failure in the model.

*The LEM model used for the stability analysis has been explained in more detail in Section 3.5. Further details, especially on the method adopted to select the shear strength parameters, have been added also in Section 4.5. It is worth to underline that results from Huang et al. (2024) have been assumed only for reference. The model proposed by Huang et al. (2024) has not been used to calculate shear strength parameters, due to the lack of experimental measurements on specific ice content. Starting from the original Mohr-Coulomb's shear strength parameters of ice, cohesion was progressively reduced by 25% for each scenario C1-C5, while the friction angle was reduced by 11% for each scenario. Anyway, the adopted values of shear strength parameters in the LEM model are consistent with those experimentally measured by Huang et al. (2024) in similar stress conditions, i.e. with stresses normal to the sliding surface in the range of 150-250 kPa. In fact, considering the unit weight of the ice equal to 9.19kN/m3 and a mean ice thickness of 20m, the mean vertical stress at the slip surface is around 184kPa. Huang et al. (2024) performed shear tests under normal stresses of 150kPa and 250kPa.*

*We for instance write in Section 3.5:*

*P4*

*"LEM is based on the principle that a rigid mass (in this case made by ice), will fail when the driving forces, due to gravity and external loads, exceed the resisting forces, due to shear strength along a defined failure surface. In the present study, the main driving force is the weight of the unstable mass, but further destabilizing actions, represented by hydrostatic forces in various configurations, were considered, as will be explained later. The calculated driving actions are compared to the available resistance, which is calculated according to Mohr-Coulomb's shear strength criterion. Referring to this shear strength criterion, specific strength parameters are assumed for the involved materials, i.e. ice and rock. The method considers the equilibrium of forces and/or moments along a predefined failure surface."*

*And later (in Section 3.5):*

*P5*

*"In the framework of LEMs, slice methods are used to analyse the stability of slopes by dividing the unstable mass into vertical slices. These methods evaluate the equilibrium of each slice while considering forces acting within and between them. Each slice is analysed separately, considering its weight, normal force, shear force, and inter-slice forces. The stability of the entire slope is determined by calculating FoS based on shear strength and equilibrium conditions."*

*And finally (in Section 3.5):*

*P6*

*"… by using the SLIDE2 software from Rocscience®. In particular, the Janbu's simplified method is based on equilibrium of slices along two orthogonal directions. It assumes only horizontal interslice forces and thus the interslice shear forces are zero (Janbu et al., 1956). The Janbu corrected FoS is obtained by multiplying the Janbu simplified FoS by a modification factor, which is function of the slope geometry and the strength parameters of the material. The Janbu modification factor is an attempt to compensate for the fact that the Janbu simplified method satisfies only force equilibrium and assumes zero interslice shear forces. Instead, GLE/Morgenstern-Price method assumes a half sine interslice force function to define interslice forces and is based on complete equilibrium of slices along two orthogonal directions and with respect to rotation. This is why this method is 'rigorous'. It should be noted that, although the failure surface was completely known, in the LEM model it was defined only at the base in contact with the rock, with the aim of comparing the failure scarp within the ice mass obtained from the model with that actually observed in situ."*

Some of the mechanisms in the model are confusing. What exactly does plasticization of the basal layer mean? First, it is not clear what the mechanism for this is (see the permafrost discussion above). Second, a 'plasticization' would lead to increased deformation rates, which should be observable. Hydraulic jacking is also a strange mechanism as applied here, because it is inconsistent with a frozen basal

condition and a plugged drainage system. How does water access the entire base of the ice under these conditions?

*This is indeed intricate yet intriguing at the same time, as basal sliding or hydraulic jacking are not possible with perfectly frozen, hydraulically impermeable basal conditions. Instability may therefore have been caused by either partially non-frozen conditions or by a frozen but partially permeable state of the basal layer and subglacial rocks. In both cases, pressurized water at the base of the large median crevasse, combined with rising temperatures, could have forced its way through the thin basal moraine, increasing permeability in the basal layer without fully reaching the outlet at the frontal toe of the glacieret. This condition may have created hydraulic jacking in the eastern portion of the sliding surface (e0 in Figure 9). On the other hand, the southern and western portions of the sliding surface are frozen to bedrock, as inferred from helicopter images taken immediately after the collapse, LIDAR surveys from the following day, satellite images from July 7th, and direct observation. These differences in the sliding surface were already highlighted in our previous study (Bondesan & Francese, 2023). For these reasons, we incorporated hydraulic jacking and the weak basal layer into the model to account for the simplified effect of friction resistance reduction (which depends on effective stress) due to increased pore water pressure in a thin layer composed of ice, water, and debris. We have also attempted to clarify these concepts in the description of how the model was conceived.*

It is fair to assess how the LEM model does, since that is used in practice. But I think a first step is to use the fact that the glacier failed to estimate what stresses actually led to failure. You know that nature and magnitude of the failed surface, and you know all the relevant stresses (gravitational stress from the weight of the failed ice body and hydrostatic stresses from the water-filled crevasse. This allows you to calculate an average stress on the failed surface, which is a good place to start.

*We understand the observation; it is possible that we were not clear enough for those unfamiliar with this approach. "Using the fact that the glacier failed to estimate what stresses actually led to failure" is exactly what we call "back analysis". The failure happened because, along the sliding surface (whose geometry is known), the driving shear stresses were higher than the resistant shear stresses. Although in a simplified way, the comparison between driving and resistance forces (or stresses) is made by using LEM and by calculating the FoS along a specific failure surface. The average stress on the failure surface is calculated in the adopted model for different scenarios and it is compared with the basal shear strength.*

It seems overall, the extremely warm temperatures and the fact that a crevasse is filled with water to the top is probably the best indicator of how failure occurred. The excess pressure from the water-filled crevasse would have led to hydraulic jacking that would have progressively increased from the crevasse downward. In that area, shear strength would be lost entirely and the stress on the remaining intact surface would increase until failure.

*Agreed with some additional detail: The subglacial freezing process following the decoupling of the glacieret from the main glacier and with it becoming a cold massive ice body may have drastically reduced the hydraulic permeability of the subglacial karstic rocks and thereby prevented the water in the*

*crevasse to subglacially drain but to build up above-floating water pressure. This development may have over-stressed the warm-soft ice and water containing permafrost bedrock at the front of the glacieret.*

The abstract is full of speculation and conjecture. A warming of subglacial permafrost is not really documented in this paper. Neither is a 'plasticization of basal ice' or the presence of a subglacial active layer.

*The Abstract has been modified (see P0 above), with most speculations removed from the text. As for subglacial permafrost, we acknowledge that it is not confirmed by direct measurements. However, given the small size of the glacieret, it is highly probable that it has degraded—at least along the perimeter—following the overall Alpine trend, as supported by 15 years of observations at the PZB site.*

*Regarding the 'plasticization of basal ice', we agree that the expression is not appropriate. Anyway, we believe that the presence of a basal layer made by ice, water, and sediments, can reasonably explain the sudden and abrupt triggering of the failure. By disregarding any kind of deformation, the weak basal layer has been introduced to take into account in a simplified way the decrease of friction resistance (which depends on effective stresses) due to the increase of pore water pressure at the contact between ice and bedrock. There are at least three key pieces of evidence supporting this hypothesis.*

*- Back-analysis results indicate that instability conditions would not have been reached without the presence of a weak basal layer.*

*- The sliding surface analysis shows that, on the eastern flank, for 32% of the entire surface, the failure developed at the ice-rock interface. This suggests that, in this area, the glacier was not frozen to its base, unlike in the western sector and around and above the median crevasse.*

*- Infrared (IR) measurements taken the morning after the collapse indirectly support this hypothesis, as they reveal skin bedrock temperatures on the eastern flank very close to the melting point.*

**Specific comments:**

The title is a bit of a mouthful; consider simplifying.

*In the title we would like to emphasize that multiple datasets are provided and analyzed, but the solution remains not-straightforward due to the complex conditions involved. We propose this new title:*

*Failure of Marmolada Glacier (Dolomites, Italy) in 2022: Materials for data-based back analyses of possible collapse mechanisms as related to recent glacio-climatic evolution and possible trigger factors.*

l.14: delete 'partially' (isolated portion and partially are redundant)

*The sentence was modified;*

l.22: ... understand THE role ...

*The sentence was modified;*

l.36/37: ice shelves are not collapsing due to acceleration and thinning of feeder glaciers, it's the other way around.

*The sentence was modified;*

l.48: neo-formation of -> formation of new ..

*The sentence was modified;*

l.48: is there really evidence for more subglacial lakes? I didn't look through the references carefully, but didn't see anything on a cursory look.

*The word "subglacial" was removed;*

l.65-70: a lot of this is unclear and speculative ('probably played a primary role', 'presumably', ...). How does active layer thickness affect glacier stability? Or are you referring to a subglacial active layer?

*The sentence was reworded (see P1 above):*

l.92: 'could be outlined' -> 'is'

*The sentence was modified;*

l.94 ; -> .

*Done.*

l.144: 'could be' -> 'was'

*The sentence was modified;*

l.161: delete 'a'

*We found the expression "As a result" to be correct;*

l.164: The hypothesis about changing plasticity appears out of thin air here. Perhaps it should be stated as a hypothesis to explore; same for active layer.

*The sentence has been modified, and the two distinct phenomena, as suggested by the reviewer, have been presented as hypotheses to be explored. The topic was also better stated in the Introduction (see also P1 above).*

Sec. 3.3: Satellite imagery and seismology is a bit of an odd combination for the same subsection

*The content was split in two sub-sections.*

l.194: define 'GSD'

*A definition was added;*

l.208: lately -> later

*The text was corrected;*

l.215: .. using a 0.05 m ...

*The text was corrected;*

l.216: define RES (or just use GPR throughout)

*RES (Radio-Echo Sounding) was already defined in the Introduction;*

l.221: functional -> used

*The text was corrected;*

l.233: by -> from

*The text was corrected;*

l.234: using values from sea ice studies seems odd; it has quite different strength

*We agree with the reviewer but we could not find other reliable data in the literature.*

l.237: 'could be probably considered': This kind of statement is very vague. Just state what exactly you assumed. Your borehole data provide justification for polythermal conditions.

*The sentence was reworded;*

l.238: Similar to what?

*The sentence was reworded;*

l.239/40: I am not convinced that PermaNet temperatures are directly indicative of polythermal glaciers. You actually explain this later; how firn refreeze processes can create warming conditions on glaciers that are not present on ice free ground and that are not incorporated in permafrost models.

*True. The sentence was removed. We now write in the Discussion:*

*P7*

*"Such recent subglacial freezing may likely have influenced the hydraulic permeability of the karstic subglacial rocks and, hence, the drainage mechanisms of the cold surface ice body."*

l.247: A bit more detail on the model is warranted here. The Supplementary Materials don't help much. Say a bit more about the model, how it works, and what the differences are between the different versions.

*The LEM model has been described in much greater detail, including the key differences between the Janbu simplified, Janbu corrected, and GLE/Morgenstern-Price methods, which have been added in Section 3.5 (see P4, P5 and P6).*

Fig. 3 caption: .. for each OF the different ...

*The text was corrected;*

Fig. 4 and elsewhere: what is the difference between 'area' and 'surface'? Is one the map area and the other one actual surface area? If so, I would stick to map area. The actual surface area of a rough surface is actually not well defined and is scale dependent.

*The difference is "map area" and "surface area" as suggested by the reviewer. We are keen to retain both graphs, as the accurate measurement of surface area over the years was a long and rather complicated task to obtain reliable results. It was based on: digitization of contour lines from available maps, digital cartography, terrestrial and airborne laser scan data, and, finally, the correlation of satellite or aerial images (see SM for details and proper references).*

l.286: precipitations -> precipitation

*The text was corrected;*

Fig. 8: the grey shaded area for Marmolada seems large, given the altitude range of the glacier

*Yes. We added in the figure caption:*

*P8*

*"The large gray scale for the Marmolada detachment site indicates the uncertainty involved with applying temperature information from different times and obtained with different methods/accuracies."*

l.337: after the mid of June -> after mid-June

*The text was corrected;*

l.339-345: several points are not clear here. You show temperature profiles for 3 July each year. These do not show the depth of the active layer. The active layer is the layer that changes from a frozen to a thawed state during the year. In mid-summer there is still part of the winter cold wave in the ground. I

don't think a single profile can be used to determine depth of the active layer. Overall, this profile does not contribute much of relevance to the paper. It shows that at a lower elevation and different exposure there maybe be some permafrost, but how transferable is this to the Marmolada Glacier with a layer of glacial ice?

*Temperature (T) data at the PZB site have been continuously collected since the borehole installation in late 2010. These data provide precise insights into how the active layer has changed over the years (2010–2025). We chose to present the data as shown in the figure to highlight the conditions in July 2022 compared to the same period in all other years. In 2022, T reached an absolute maximum within the depth interval between the surface and 3.5 m. These data offer some insights into how the thermal wave extended in depth in 2022 compared to the previous decade. Additionally, analyzing the PZB dataset allows for a local validation of the permafrost distribution map proposed by Boeckli et al. (2012) and, consequently, an assessment of its applicability to the Marmolada site.*

*We now write at Section 3.2:*

*P9*

*"The model devised by Boeckli (2012) is based on Mean Annual Air Temperature (MAAT) of the period 1960-1990. PZB data confirm this model and provide some information about ongoing permafrost warming in the region."*

l.358/59: I'm having trouble identifying the 'traversel bediere' in the figure. This seems like a real oddity: how could a stream of water run in a transverse direction on a steep glacier?

*We understand reviewer's observations. What we are trying to describe is not a bédière strictu sensu but rather a crevasse filled with water, where the transverse water flow erodes the ice, creating a typical meandering morphology. The flow is transverse because the lower face of the crevasse forms a counter-slope, preventing the water from flowing in the longitudinal direction. At the end of the transverse section, the water overflows from the meander and flows over the ice surface, creating a true bédière along the direction of maximum slope. We replaced "bédière " with "small torrent" and we added some explanation to the text and figure caption.*

l.381: 'Very little water evidences are' -> 'Very little evidence for water is'

*The text was corrected;*

l.382: delete 'somehow'

*The text was corrected;*

Fig. 11: I'm a bit skeptical about radar interpretations of bedrock under ice. These radar profiles are most likely influenced by out-of-plane reflections due to the shape of the glacieret. The v-shaped

troughs indicate that they should probably be migrated. The interpreted lines in the bedrock look quite a bit like the tails of hyperbolae created by point reflectors. Migration would shed some light on that.

*The authors are aware that out-of-plane reflections may occur in this geometry, with the crevasse signature c1 being a typical example. We collected the data using high-frequency (500 MHz) shielded antennas to partially mitigate this phenomenon in the bedrock response. The radius of the first Fresnel zone for this antenna frequency is approximately 1.5 m at a depth of about 15 m (the maximum depth of the bedrock), thus limiting the footprint size and the extent of backscattering from buried reflectors. We also recognize that some hyperbola tails are likely due to peaks and troughs in the bedrock morphology. However, as a first approach, we chose not to migrate the section to avoid the typical smearing of the radar image and enhance interpretation of the weak near-surface reflectors. Nevertheless, the post-failure radar profile was migrated. The new Figure 12 now shows both the un-migrated and migrated images of the bedrock. The bedrock geometry was reinterpreted based on the migrated data, resulting in a maximum vertical-lateral shift of 1 m. This re-interpretation led to a smoother representation of the bedrock morphology and it was validated comparing the buried bedrock morphology with the shape of the outcropping rocks nowadays visible just below the failure and in addition the maximum depth reached by the borehole in the ice nicely correlates with the RES interpretation in that position of the profile.*

Fig. 12: what's the colorscale?

*The color scale is "the normal" and it is commonly used to enhance surface orientation. RGB colors encode changes in surface normals.*

l.416: correct m3

*The text was corrected;*

l.418: could be -> was

*The text was corrected;*

l.421-24: can you conclude anything from these temperature measurements? This was imagery in the early morning and mostly reflects 'skin temperature', which would be very influenced by air temperature over the night.

*These temperature measurements always taken in the early morning are indeed skin temperatures. However, the values are consistent and comparable, despite having been collected in different seasons (summer and autumn). Furthermore, they are much lower than the minimum air temperature recorded during the previous night. At least in a qualitative sense, this information confirms that the ice body is cold. The above comment was added to the text.*

l.425-427: the surface temperature cannot be used to say something about permafrost conditions.

*We agree with the reviewer and have reprocessed the temperature of the exposed rock measured on the morning after the collapse, which ranged between 0 and -1°C. This temperature indicates thermal conditions close to the melting point (although influenced by nighttime air temperatures) and therefore could supports the hypothesis of ice/sediment softening at the basal level over approximately 32% of the failure surface.*

Fig. 14: The seismic record is interesting. How well is the timing of the avalanche known? For example it looks like the seismic event records the initial failure (mostly in the horizontal components) and then the impact of the falling ice with both vertical and horizontal components about 20 sec later? Is this a reasonable interpretation?

*The error in the timing of the ice avalanche is approximately 0.3 s (0.27 s to be more precise) and it was estimated using multiple seismic stations. The reviewer's analysis is, in principle, correct but we are dealing with a rather complex wavetrain. The initial failure is primarily recorded in the horizontal component, followed by an ensemble of body (mostly S), converted and surface waves. The impact of the falling ice occurs on surface with an inclination of approximately 30 degrees thus generating high-amplitude transversal waves as well as lower-amplitude longitudinal waves. The time difference (Δt) between these two events (initial movement and impact of the falling ice) was recorded more than 20 km away from the glacier. The differing velocities of body and surface waves, along with dispersion, cause this Δt to become larger as the recording distance increases.*

l.460-464: I fail to see the relevance of this paragraph and Fig. 14B. First, the figure shows no obvious similarity between the two events (for example the Everest event had a much larger vertical component). A thorough analysis would require looking at spectra and perhaps some force-momentum modeling.

*The paragraph was removed and the spectra of the Marmolada failure were included in the figure. A brief comment of the frequency content of the different spectral components was also added.*

Fig 15 caption: what does 'with evidenced the lower transversal crevasses' mean?

*The text was taken out for clarity*

There is a lot of overlap in Figs 15-18; these could be consolidated.

*The figure showing the schematization (conceptualization) of the numerical modeling was removed.*

l.510: what do you mean by 'uneven combination'?

*The text was changed in "unfortunate combination";*

l.518-520: Here is where you state your main hypothesis for the thermal state, but this is only weakly backed up (see discussion on permafrost above).

*The discussion regarding the thermal state has been simplified and re-organized to better align with the available data, avoiding hypotheses and speculations*

l.526: snow precipitation is mentioned here, but it is not well motivated.

*The sentence was expanded and now it reads:*

*P10*

*"The longer-term conditioning/predisposing factors are mostly determined by the relationships between the geometry of the glacieret and that of the carbonate bedrock. Further predisposing factors depend on the internal structure of the glacieret itself, on the physics and stratigraphy of the basal layers, the permeability of the frozen sub- and peri-glacial glacial rocks, and on the amount of winter-spring snow precipitation, which affects the timing of glacier ice exposure during the summer."*

l.529-30: this is very much a hypothesis and conjecture and would require some thermal modeling to make a more definitive statement.

*The sentence was removed to avoid conjectures. Thermal modeling is a good idea for future work.*

l.544: second tens of -> middle of

*The text was corrected;*

l.556/7: Clarify what you mean here when you talk of 'thermal inertia in the exposed ice body'.

*The sentence was removed for clarity purposes;*

l.570: pervious -> previous

*In this sentence "pervious" was used as a synonymous of "permeable";*

l.577-85: Again, there is quite a bit of speculation here

*The text describing the impact of permafrost changes on glacier stability was removed to avoid speculation;*

l.629: Altel -> Altels

*The text was corrected;*

l.634: Finding temperate ice with RES is a fraught subject with often questionable conclusions.

*The sentence was removed;*

l.664/5: Again, very speculative

*The part of the sentence referring to the 'transfer of heating from the outcropping rocks forming the east shoulder' was removed. The sentence was reworded to avoid hypotheses about the formation of the weak layer, instead focusing on describing the nature of the weak basal layer as reported in the literature.*

l.673/4: ditto

*The sentence was removed;*

l.711/2: this is an interesting observation and it needs some elaboration. It is not clear how a crevasse could penetrate to near the base in a cold glacier without the help of water. If the glacier is frozen to the bed, there are no longitudinal strain rates there, and failure under tension is not possible.

*The sentence has been modified and expanded to invoke the action of water as one of the agents for the opening of the crevasse.*

l.740: something is missing here.

*The sentence was reworded;*

The data statement reads odd. Every effort should be made to make data available publicly. That may not be possible with some of the proprietary imagery. But statements like 'data could be made available upon request' do definitely not meet modern open data requirements of most journals.

*The reason for this unusual statement is that we are dealing with a variety of datasets collected over the years from multiple public agencies, foundations, non-profit organizations, and private corporations. The authors can provide access to the geophysical data they collected after the failure, including RES profiles, IR images, and temperature measurements from the borehole drilled in the ice, as well as the seismological data. Additionally, they can share the DTMs generated by processing scanned historical maps, digital maps, and aerial/satellite images. The meteorological time series (temperature, snow cover, and rainfall) are publicly available, but a request must be submitted to ARPAV (https://www.arpa.veneto.it/datirete.htm/richiesta-dati). Finally, pre- and post-failure satellite images, which were purchased by the authors, cannot be made available.*

---

## Author Comment (AC2)

**Reply to reviewer #2 (Anonymous reviewer)**

First, I would like to commend the authors for compiling extensive data related to the tragic collapse of the Marmolada Glacier. They have gathered information from multiple sources to evaluate the factors that triggered this catastrophic event. This manuscript is undoubtedly valuable and has the potential for high citation and recognition within the scientific community.

*We thank reviewer #2 for the constructive comments.*

However, before publication, the manuscript requires major revisions. This includes shortening and restructuring the content, as well as thoroughly checking for numerous redundancies. Additionally, the discussion of certain processes related to the glacier's thermal regime and permafrost needs re-assessment, as also addressed by M. Truffer in Review 1. While I started reading with eagerness to learn more about the event, I was somewhat disappointed because of obvious shortages. In my opinion, the manuscript is overly lengthy, the figures are not cited in order, and some are redundant and difficult to interpret. Below, I provide major and detailed comments. I have read M. Truffer's comments and fully support the points made, especially regarding the glacier's thermal regime and permafrost as possible triggers for the landslide.

*The paper has been partially re-organized. The content was shortened and restructured and we worked on redundancies as well. The "Introduction" Section was shortened and the objectives clearly stated as well as the key methods. The "General Settings" Section was expanded adding geological and geomorphological settings. We carefully re-assessed many of our statements about the glacier's thermal regime (see the various replies to the review of Martin Truffer that's not copied here to avoid duplications). All figures were reworked to improve clarity and they are cited sequentially. The font size was increased, and some panels were removed. Panel C (graph from Davies et al.) was removed from Figure 8. Figure 11, which includes radar profiles and their interpretation, was split into two figures. The time series from the Himalaya avalanche were removed, while the spectral components of the Marmolada failure were added. Figure 15 was simplified (removing one cross-section) and Figure 16 was removed.*

**Specific comments:**

**Abstract**: Move the sentence starting at line 17 to the end, after listing the methods used, as it is repeated there. Also, clarify whether the active layer in the permafrost is below the glacier, we do not have any "active layers" below ice.

*The abstract was rewritten (see point P0 of our reply to Martin Truffer's at page i).*

**Introduction**: This section is too long. Remove the initial list of collapses, including the ice shelf, as it is not relevant. Delete everything from line 65 onward related to "overall structure," as these discussions belong later. The introduction should end by stating the objectives of the paper and mentioning key methods (e.g., climate observations, thermal regime, seismicity) used to achieve these goals.

*The Introduction was shortened and re-organized. The list of collapses was summarized. The objectives of the paper were clearly stated as well as our claim. The sentence stating the objectives now reads:*

*P1*

*"This study aims to partially address this gap with a dual objective: first, to identify the controlling factors of the collapse and classify them as predisposing or triggering; and second, to develop a numerical model to assess the relative influence of the different triggering factors. Multiple variables contributed to the catastrophic collapse that claimed several lives on July 3, 2022 (Chiarle et al., 2022; Olivieri and Bettanini, 2022; Bondesan and Francese, 2023). This required the collection and reorganization of various data into a comprehensive digital database to gain deeper insight."*

*We also reformulated the sentence indicating the focus of the numerical modeling that now reads:*

*P2*

*"We used a data-based back-analysis approach to infer the basal properties of the failure surface, aiming to understand the critical interactions among englacial water (which altered temperature and pressure fields within the glacier and at its base), permafrost in rocks and sediments, the glacier's thermal state, and the possible presence of a thin, heterogeneous, and discontinuous layer at the ice-bedrock interface."*

*And we better summarized the key methods:*

*P3*

*"Numerical simulations were conducted by means of the Limit Equilibrium Method (LEM), which is routinely used for slope stability analyses (Saim and Kasa, 2023) in geotechnical engineering. Particular attention was given in defining geometry and physical properties of the ice body, especially for those characterizing the interactions with the surrounding materials at the ice-rock interface. The purpose was achieved by re-processing and carefully analysing both existing and post-failure RES (Radio-Echo Sounding) profiles (Fretwell et al., 2013; Francese et al., 2019), which contributed to the conceptualization of the model for numerical simulations. Pre- and post-failure aerial and satellite imagery, along with aerial and terrestrial laser data, further contributed to conceiving the model. Available meteorological data (air temperature, rainfall, and snow cover) and cryospheric data (permafrost and ice temperature) were carefully analyzed. Finally, seismological observations were considered to evaluate the possibility of earthquake-induced triggering of the failure.*

*We prefer to leave the paper summary at the end of the "Introduction" as it helps the reader.*

**General Setting:** The field area is not adequately introduced. Provide information about the overall geological and structural setting, climate (temperature and precipitation), and permafrost limits in the area. In Figure 1, the text is too small, there's no scale in (A), and the 3D plot's scale is not readable. It would help to include a regular image of the area and images from the collapse.

*The geology, geomorphology, and climate of the study site have been described in greater detail. The font size in Figure 1 has been increased, and the scale bars have been enlarged for better readability. The scale in subpanel A was originally in degrees and minutes, which may have been difficult to interpret, so we enlarged its size. Additionally, a regular image has been included to replace the 3D model in Figure 2.*

**Data, Methods, etc.**: Remove the initial paragraph listing methods, as these should be detailed with manufacturer information, resolutions, and other relevant details. Line 159 contains an interpretation that repeats throughout the manuscript; avoid such redundancies. The calculation starting from line 169 is unclear; specify if GPR was used and the strategy for filling in missing meteorological data. Figures are cited out of order (e.g., Fig. 9 on line 189), so ensure they follow a sequence. Important installations could be visualized in Fig. 1 or 2; consider using a standard map layout instead of complex 3D plots. I found it challenging to follow the glacier stability and back analysis. The discussion of parameters is confusing and should be reserved for the discussion section—just present the parameters used along with references and justification.

*The list has been removed, leaving only the first introductory lines of the section, referring to the following paragraphs for the description of the available data and the methods used. We also removed the interpretation provided in the subsequent lines to avoid repetitions. Now the incipit of Section 3 reads:*

*P4*

*"The multi-disciplinary approach of the present study involves several research topics and different data types along with the specific methods of data acquisition, analysis and processing. Data were gathered in a large database comprising a wide range of glaciological and meteorological records, along with historical and modern topographic maps, aerial and satellite imagery, geophysical data, geological and geomorphological data collected and catalogued over the past two decades (see Supplementary Material for details)."*

*The calculation was better described, and the strategy for filling in missing meteorological data is outlined in the Supplementary Material and our previous paper; proper references have been added to the text. Figures are now cited in the proper order. Figures 1 and 2 were improved in both graphics and labeling, and the installations are now clearly visible. The theory of the back-analysis via the LEM approach was expanded (see reply to Martin Truffer). Parameters are now simply presented, with other considerations on selection criteria were shifted to the "Discussion" section as requested.*

The ice temperature of -4°C seems low; is this value justified?

*-4°C is a lower-bound value. We provide a best estimate of about -2°C and in Figure 8 a range from close to 0°C to -4°C. In the caption to figure 8, we write:*

*P5*

*"The large gray scale for the Marmolada detachment site indicates the uncertainty involved with applying temperature information from different times and obtained with different methods/accuracies."*

**Results**: What is meant by "overall evolution"? Call it "development" or just "evolution." It is not interesting here who conducted the measurement (l. 251), just provide the results. Increase the font size in Fig. 3, as it is hardly readable. Fig. 4c is difficult to understand, and I cannot read the figure. What is meant by the "centroid of the glacier front"? Low readability also applies to Fig. 5—text within the figure is not clear. Ensure figures are readable on paper. I gave up on Fig. 6 because, while it looks nice, it is not understandable.

*We changed "overall evolution" in "evolution" and removed the indications on who conducted the measurements. Font size has been increased in all figures (see above). Figure 4C represents the spatial migration of the centroid of the glacier front and its projection onto the EN, EZ, and NZ planes. The idea behind using the centroid is to provide a more robust representation of how the glacier front is changing over time, thus overcoming the limitations of describing the glacier front migration with a few sparse measurements taken at specific spots. The digitized 3D polyline representing the glacier front was first approximated with a 3D spline and then collapsed onto a single point (x, y, z), providing a much more accurate representation of the glacier front migration (this process is described in the Supplementary Material of Bondesan and Francese, 2023). A brief explanation and a reference were added to the caption.*

Figure 6c presents a curve done for two cement blocks, not for the ice-bedrock interface. There's no certainty that the same relation is valid here; consider removing it. Furthermore, what is meant by "guessed thermal conditions"?

*Panel C (graph from Davies et al., 2001) was eliminated from Figure 6. The term "guessed thermal conditions" was changed in "thermal conditions".*

Line 346 and following contain interpretation, not results; this should be moved.

*The sentence was moved to the "Discussion".*

Chapter 4.3. is difficult to follow. Figs. 9 and 10 are very challenging; consider using color and clear color ranges. These are certainly important figures, but difficult to follow.

*This description of the failure zone is a core component of this paper, as it forms the foundation for conceptualizing the numerical model used to assess stability and evaluate the different triggering factors. The validity of the model is strictly dependent on the settings of the failure zone. We attempted*

*to clarify the description by leaving out some of the details and focusing on the key issues. Additionally, we improved the line drawing overlay on the image by using wider lines and different colors for better clarity and distinction. These pre- and post-failure satellite imagery was provided in standard and widely accepted colors. RGB color-coding of the images does not enhance readability, and grayscale proved to be the most effective display choice.*

Fig. 11 is similar (especially parts A and D). What is RES? What does "very short wavelength" mean in line 400?

*Figure 11 was split in two figures to improve readability. The acronym RES was already expanded in Radio-Echo Sounding in the "Introduction" Section. "Very short wavelength" is a common term in geophysics, indicating higher resolving capability. In wavenumber methods, the shorter the wavelength, the higher the spatial resolution.*

Fig. 12 is very difficult to read. Line 425 should be revised, as surface temperature cannot be used to infer permafrost under the conditions mentioned.

*Line drawing and labeling in Figure 12 were improved increasing the width of the lines and the size of the fonts. The sentence in line 425 has been completely revised and incorporated into the new discussion on subglacial permafrost. Please refer to our response to Martin Truffer's comment "A warming of subglacial permafrost is not really documented in this paper" on page vi of this document.*

This chapter about seismology looks good, including the figures.

*Ok; We included comments on the spectra of the failure event.*

**Slope Stability Back Analysis**: I found this section difficult to follow—perhaps it is my fault—but the figures (e.g., Fig. 15) are not user-friendly.

*The conceptualization was simplified as well as Figure 15 that now shows a single longitudinal cross-section to be used as a geometrical reference for the model.  Figure 16 was removed. Figure 17 and Figure 18 are common representations of failure surfaces calculated according to Mohr-Coulomb's shear strength criterion.*

Discussion: There is much redundancy here. The first paragraph provides a conclusion rather than a discussion. Section 5.1. is repetitive (line 574). How do you know the thermal regime of the LIA glacier (line 565)? Some observations have been mentioned before and seem redundant. The section on "triggering factors" is very lengthy; consider creating subheadings like "seismic factors," "thermal factors," etc.

*The 'Discussion' section was reorganized and divided into more clearly defined subsections. Some sentences were removed to avoid redundancy, while others were relocated to different subsections for conciseness. Several statements were streamlined to enhance readability, and the first sentence was*

*moved to the 'Conclusion' section. The discussion on permafrost was simplified (see P3 of our response to Martin Truffer's comments on page iii). Subheadings were added to Section 5.2 (Triggering Factors), and the section was shortened.*

Conclusions: The conclusions contain much speculation and a few typos around line 740. I recommend using short conclusions with clear statements in bullet points rather than long text. You can omit the last paragraph, as it is not informative.

*We have shortened the "Conclusions" Section avoiding speculations. We included just clear statements and removed the last paragraph. Now the core of the "Conclusions" Section reads:*

*P6*

*The detachment zone, as a consequence of the warming-induced disaggregation of the Little Ice Age glacier became an isolated cold glacieret—consisting of massive, impermeable, yet crevassed ice—at least over the past three decades. The progressive opening of a large median crevasse increased the glacieret capability of storing englacial water. Its ice temperature can be estimated at some -2°C, i.e., relatively close to melting conditions but with at least partially freezing or frozen bed. Additional conditioning factors included the steep slope inclination, the presence of low-angle discontinuities such as ice foliations and/or discontinuous basal till layers, along with the complex thermal conditions at the ice-bedrock interface.*

*The probably most influential triggering factors are associated with minimal winter snowfall and the prolonged positive thermal anomaly. The marginal thickness of low-permeable snow layers and especially the extreme air temperatures resulted in an excess of meltwater penetrating deep into the glacier. In fact, water filling the deep crevasses have produced subglacial water pressures in excess of floating conditions. The absence of a connected drainage network created the condition for the development of an increasing hydraulic over-pressure.*

*An earthquake, as the final triggering mechanism, can be excluded. Results from numerical simulations suggest that the triggering of the final collapse was most likely due to the simultaneous interaction of hydrostatic pressure, hydraulic jacking pressure, and a reduction in basal friction caused by the presence of a weak basal layer. This thin layer appears to be a key factor in the failure, as instability does not occur by hydraulic pressure alone. Its evolution is still under investigation, but it may be correlated with permafrost degradation at the ice-bedrock interface, leading to partial basal ice melting across the entire eastern flank of the glacieret's base.*

---

## Author Response (AR2)

**Reply to Dr. Truffer**

This is a second review of the paper. The authors have made significant revisions and the paper has become much clearer and more focused in the process. I do recommend publication, but I still have a few issues that should be clarified.

*The authors are grateful to Martin Truffer for undertaking this second round of review.*

**General comments**

Mostly, those revolve around the discussion of the role of permafrost that is, at times, still confusing. The paper claims on the one hand that glacier recession led from a temperate bed condition to a freezing bed condition since the LIA, and then more recently to a warming of the permafrost. I find that the paper presents little evidence for that. In fact, the authors present some evidence that during World War 1 the basal temperatures in the failure zone were below freezing. Borehole measurements immediately following the failure also show below freezing conditions.

*We concur that the overall evolution of the thermal conditions cannot be definitively established from direct evidences, however, we present a logical and internally consistent reconstruction of the governing processes. The Little Ice Age glacier was most likely "polythermal" and predominantly "warm-based" except for the cold margins with underlying subglacial and adjacent proglacial permafrost.*

*In a steady-state condition, the summer meltwater percolates into the permeable firn of the accumulation area.*

[Figure]

*Subsequent refreezing of this meltwater within the firn releases latent heat, which warms the firn matrix and maintains it at the pressure-melting point (i.e., at phase-equilibrium temperature). This internal*

*refreezing process ensures that a significant portion of the energy absorbed at the surface, leading to melt, is retained as heat within the firn layer of the accumulation area.*

*The temperate ice produced by the snow-firn-ice metamorphosis flows down underneath the equilibrium line to the ablation area. As the ice of the ablation area is impermeable, the meltwater flows off at the surface and takes with it the energy input from surface melting. The ice below the equilibrium line therefore cools down under the influence of negative surface temperatures forming an increasingly thick wedge of cold ice, which reaches the glacier bed near the ice margin, where permafrost penetrates into the ground. The Gruben glacier in the Swiss Alps provides a well-documented example of these conditions, and historical temperature records from World War I are consistent with such an incipient cooling stage*

*Global warming perturbs these equilibrium conditions. The equilibrium line rises rapidly, while the glacier typically retreats more slowly, reflecting its thermal and dynamic inertia. As a consequence, the warm firn area progressively diminishes and may eventually disappear completely. The glacier thereby transforms into an ice body situated entirely within the ablation regime. Influenced by recurrent sub-freezing ambient temperatures, this formerly temperate ice now cools and becomes cold ice. Where this cold ice is in contact with the bed, subglacial permafrost can begin to aggrade. This process tends to stabilize remaining glacier parts.*

*When the glacier finally disappears, the now exposed former glacier bed become perennially frozen , exhibiting newly formed permafrost. This is likely the current situation in deglaciated parts of the Marmolada massif, where bedrock has been exposed by glacier. Global warming, however, continues affecting permafrost worldwide. This means that the newly formed permafrost in the recently deglaciated areas including those surrounding our remaining glacieret, is also subject to warming.*

[Figure]

*In addition, meltwater infiltrating crevasses can induce refreezing (regelation) processes and convective heat transfer, contributing to the warming of sections of the glacier base.  These factors, in conjunction*

*with associated hydraulic pressures, can affect the overall stability of the glacier —especially at the margins.*

*We changed the sentences in section 5.1.3 and now they read:*

*P0*

*"Evolution of thermal conditions around the failure zone is crucial to get better insight into potential collapse mechanisms but involves complex transient glacier-permafrost interactions (Figs. 7-8). Where bedrock of the northern slope has remained free of glaciers during past decades, its temperature can be estimated at a few degrees Celsius below freezing, ranging from typically near 0°C at lower altitudes, around 2400-2600 m asl, and reaching -1°C to -4°C towards the uppermost parts. With such surface temperatures, permafrost depth may in places exceed several tens of m (Etzelmüller et al., 2020). In the Alps, permeable firn zones, warmed up by percolating and refreezing meltwater, are temperate up to altitudes between approximately 3400 and 3900 m above sea level (Haeberli and Alean 1985; Suter et al., 2001; Bohleber, 2019) while the impermeable ice of the ablation zones in areas with permafrost tends to be cold. The LIA glacier at the Marmolada site with extensive warm firn areas may therefore have been polythermal and predominantly warm-based (cf., Wilson and Flowers, 2013) with only its lowest margins containing cold ice, partially frozen to bedrock. Higher parts of bedrock underneath the LIA glacier could therefore have remained largely unfrozen. Interestingly, ice temperatures recorded several tens of cm within the walls of some WWI tunnels, dug in the Marmolada glacier in 1917 (Hess, 1940), in the elevation range 2800-3200 m asl, already showed average values of -1.32°C in tunnel "32" (located in the vicinity of the failure site) and of -1.27°C in tunnel "S" (located ~700 m west of the failure site. These values suggest that moderately cold conditions prevailed at that time on the northern side of the Marmolada. Over the following decades, the progressive loss of insulating warm firn areas likely led to further cooling of surface ice. This cooling, in turn, is thought to have affected the thermal regime of the ice body from which the avalanche later detached."*

*"The long-term thermal evolution of the glacieret has, on one hand, caused the freezing of its ice base to the underlying rock, as confirmed by temperature measurements taken in the ice borehole throughout the upper portion last summer. On the other hand, the warming-induced permafrost degradation (Gruber and Haeberli, 2007), suggested by the fifteen-year temperature and active layer and permafrost thickness monitoring in the rock borehole at PZB (Fig. 7), is likely a contributing factor to weakening the basal interface in the frontal zone of the glacier. The curved geometry of the median crevasse, clearly visible after the failure, suggests that the glacier was primarily deformed under the influence of gravitational forces, with negligible basal sliding. This further supports the hypothesis of a locally cold ice base. Moreover, as clearly shown in Fig. 9B, a large portion of basal ice remained frozen to bedrock after the failure (see more details in subsection 5.2.2). Such recent subglacial freezing may likely have influenced the hydraulic permeability of the karstic subglacial rocks, potentially reducing drainage capacity beneath the cold ice. This permeability reduction could have helped building up high water pressures in the ice body."*

To me, the most pertinent observations are the fully water-filled crevasse that would have created overpressure at the base of the crevasse, and the fact that no subglacial drainage occurred, which is consistent with a cold bed condition. Under such circumstances it should be expected that water pressure can progressively force a gap at the bottom of the glacier, leading to hydraulic jacking. The modeling section is useful in that it shows that the crevasse pressure cannot be considered as a sole independent cause of the failure. This is a significant result. One thing that is not clear to me, however, is how hydraulic jacking is considered as a separate factor from decreased ice-bedrock friction. If hydraulic jacking occurs, the friction would have to go to essentially zero.

*Hydraulic jacking and the decrease in friction along the failure surface are considered separately due to the impossibility of otherwise incorporating the principle of effective stresses into the simplified model. This principle, enunciated by Terzaghi in 1922 and later in 1947, states that the mechanical behavior (including shear strength) of a granular material (e.g. sand) submerged under water pressure depends only on the effective stresses, which are equivalent to the total stresses minus the value of the water pressure. In other words, the Coulomb-like frictional behavior depends on the inter-particle contact forces that are only diminished and not totally cancelled out by the water pressure. Therefore, in the slope stability model reported in the article, the pore water pressure is simulated by hydraulic jacking, while the shear strength is expressed through the Coulomb criterion in terms of total stresses (i.e. the decreased ice-bedrock friction), assuming the principle of superposition of effects to be valid, without calculating the effective stresses along the failure surface.*

*Tergazhi K. Der grundbruch an stauwerken und seine verhiltung. Die Wasserkraft 17 (1922) 687445–449.*

*Terzaghi, K., Theoretical Soil Mechanics, John Wiley and Sons, New York (1943) ISBN 0-471-85305-4.*

How about the following model that seems consistent with all the data and follows along similar lines as the one presented in the paper:

1) The glacier base in the failure area was frozen, presumably for many decades. This frozen condition prevents subglacial water drainage.

2) The opening of a full-depth crevasse allowed water to get to the base of the ice and over-pressure the system, progressively forming a gap that led to hydraulic jacking and a loss of basal friction.

3) General climate warming is leading to a general degradation of permafrost in the area that may have led to a warming of the ice base near the frontal area of the glacieret. This is a plausible contributing factor and might help explain why the crevasse filling event in 2003 did not lead to a failure.

*We fully agree with the reviewer, and to some extent, similar considerations were already present in various section of the paper. We have now better conveyed this timeline by adding a sentence to the Conclusion section:*

*P1*

*In short, the base of the cold ice in the failure area was most probably frozen, presumably for many decades already. This frozen condition largely, if not entirely, prevented subglacial water drainage. The recent opening of a full-depth crevasse allowed water to get to the base of the ice and to over-pressure the system, progressively forming a gap that led to hydraulic jacking and a loss of basal friction. Continued atmospheric warming leading to a general degradation of permafrost in the area may have led to a warming of the ice base near the frontal and marginal area of the glacieret. This plausible contributing factor might help explain why the crevasse filling event in 2003 did not lead to a failure.*

**Specific comments:**

l.37: ..., such as the degradation ...

*The sentence was modified;*

l.49: treats -> threats

*Correction was made;*

l.201: delete initials in reference

*Correction was made;*

Figure 3 caption: delete 'surface ablation' (none of these curves show surface ablation)

*We replaced 'surface ablation' with 'surface lowering'; Panel C shows the lowering of the ice surface over time at the failure site … in my view this is still ablation but the word was changed.*

l.269: explain the difference between 'surface' and 'area'. This was not obvious to me when I first reviewed the manuscript. I still think that reporting a 'surface' does not add additional information, and is, in fact, a scale dependent quantity, with no clearly defined meaning.

*In general, we agree with the reviewer. The reason we included surface area in the graphs is that, in the case of dipping slopes, it can serve as an additional parameter to describe changes. This quantity was also required to compute ice volume changes over time, so it was already available and we decided to include it in the figure. The Topographic Surface Area (of a slope) is the actual two-dimensional measure of the slope's terrain surface, accounting for all inclinations and undulations. This definition was added to the caption.*

l.276: a newer reference, such as Hugonnet et al may be more appropriate here

*The reference was added.*

l.292: delete 'as'

*'as' was removed and the first part of the caption was reworded.*

l.340-42: Rewrite sentence for clarity and correct grammar. Perhaps: Despite the uncertainty about the permafrost conditions at the Marmolada detachment site, we estimate the near surface temperature to be colder than at PBZ (at about -2 deg C), due to its northerly exposure.

*The sentence was changed according to reviewer's suggestion.*

Fig. 9 caption and elsewhere: how do you know this is regelation ice?

*Primarily because it is a cryoconite-enriched layer. Evidence supporting this interpretation includes the following observations: (i) cryoconite was observed to be deposited at the bottom of the crevasses; (ii) the associated ice surface was evidently distinctly glassy, a typical characteristic of regelation ice; and (iii) this dark and thin layer mantled the crevasse floor.*

l.437-444: I suppose it's ok to report these measurements, but it's hard for me to see how they should be interpreted. For example, on a clear night with strong radiative cooling, I would expect the surface temperature to be lower than the air temperature. But you are not really claiming that the ice temperature is -9 deg C, or is that the point? That would be significantly colder than any other direct measurement.

*We understand the reviewer's observations. These IR data were repeatedly collected using drone-based high-end sensors, and I also personally took some IR shots from a helicopter while hovering just above the ground and directly in front of the ice scarp. The results from the different surveys were consistent and comparable, and in my opinion, for these reasons, it is worthwhile to report these data. We are NOT claiming that the ice is at –9 °C; we are simply suggesting that it is likely a few degrees below the freezing point, as recently confirmed by borehole temperature measurements. In the paper by Aubry-Wake et al. (2015), the authors report a negative bias exceeding 5 °C.*

*We changed the sentence that now reads:*

*P2*

*These temperatures, always taken in the early morning, represent skin temperatures, but they are significantly lower than the nighttime air temperatures. The values are relevant because they have remained consistent and comparable over time. Reliable quantitative interpretation requires complex calibration of the sensor along with specific processing. Corrections on the order of several degrees are often necessary (Aubry-Wake et al., 2015) especially in the case of rough surfaces. Despite potential biases of several degrees, these measurements still indicate —at least qualitatively—that the ice body is unquestionably cold, with temperatures several degrees below freezing.*

l.442: complicate -> complicated

*Correction was made;*

l.442: delete 'Anyway'

*Correction was made;*

l.443: because THEY remain ...

*Correction was made;*

Fig. 16 caption: rewrite (I don't understand 'with evidenced the cross section'

*The first part of the caption was changed in: Pre-failure model of the glacieret showing cross-sections HH', MM', and NN'.*

l.553: what is the decimeter scale based on (estimated from what?)

*Direct observation … the sentence was modified accordingly.*

l.554: ... was performed along a cross-section that ROUGHLY follows ...

*Correction was made;*

l.539-542: I don't understand most of this sentence, including what you mean by 'heat waves propagating through the exposed rocks'. If you mean from the southern slopes on the other side, than this would imply longer time scales (many years)

*This means that it's sometimes hard to distinguish between conditioning and triggering factors, as a predisposing factor can turn into a trigger over time—especially in the case of thermal effects. We do NOT refer to long-term heating effects from the southern face of the Marmolada; rather, we are talking about solar radiation directly warming the exposed bedrock at the front and on the eastern side of the glacieret. We change the sentence that now reads:*

*P3*

*It is important to emphasize that certain thermal effects can act as both predisposing and triggering factors. In particular, dynamic thermal processes—such as heat transported to the glacier base by meltwater percolating through crevasses, and heat absorbed by exposed rock surfaces—tend to function more as triggers than as underlying conditions.*

l.546: delete 'glacial'

*Correction was made;*

l.569: sum -> total

*Correction was made;*

l.574: delete 'a' in front of precipitation

*Correction was made;*

l.576: I don't believe decade is used like this, 'in the second third of May'?

*Correction was made;*

l.577: ... absorbs less solar radiation THAN ICE, but ...

*Correction was made;*

l.577/78: is this correct in terms of mass of liquid water produced?

*We found this a reasonable estimate;*

l.580 there is a very -> there was very

*Correction was made;*

l.580: penetrate right into -> penetrates into

*Correction was made;*

l.582: is -> being

*Correction was made;*

l.584: in -> into

*Correction was made;*

l.589: do you mean 'permeable' (instead of pervious)?

*We replaced pervious with permeable although the two words are often used as synonyms.*

l.597-599: the sentence is not quite grammatically correct. But see also my overall comment. This statement is plausible, but it is not backed up by the observation of cold ice during WWI.

*We have addressed this point in the overall comments.*

*In our opinion, these temperatures indicate that the ice body is progressively cooling; leading to the cold ice we measured last summer. This is consistent with our hypothesis about the overall glacier evolution. The sentence was changed to include these considerations (see new sentence P0 in the overall comments).*

l.600-601: There is no evidence given for this statement

*What we describe here is the typical behavior observed in other Alpine glaciers when the firn layer disappears. We have addressed this point in the overall comments.*

l.603: I would state this as: ... is likely (or plausibly) a contributing factor to weakening the basal interface in the frontal zone of the glacier.

*The sentence was changed accordingly.*

l.605: I'm not sure if you could really conclude no basal sliding from that

*This is an additional indication that basal sliding was absent or negligible. Further supporting elements include: the borehole drilled in 2024 just a few meters above the failure zone confirmed that the ice was frozen to the bedrock; aerial and satellite imagery from the past 30 years shows no evidence of sliding; and no water was observed at the glacier base, as indicated by the lack of frontal discharge between 1990 and 2022.*

l.607: no evidence is provided for 'recent basal freezing'

*See our response to the reviewer's comments on Fig. 9. The presence of a thin, cryoconite-enriched ice layer at the bottom of the crevasse provides evidence of regelation ice forming at the glacier base, directly above the bedrock.*

l.614: reformulate this. I don't know what is meant by 'initial sliding of critical glacieret block thus triggering ...'

*The sentence was reworded and now it reads:*

*P4*

*Among the possible triggers, we also evaluated whether a small earthquake could have caused the displacement of an ice mass that was already near its stability threshold, thereby leading to the full failure (Figs. 14-15).*

l.621: ',' after before

*Correction was made;*

l.622: ... glacier on the day ...

*Correction was made;*

l.646: melting -> melt

*Correction was made;*

l.651: .. may ALSO have been …

*Correction was made;*

l.651-53: There is no evidence for melting in the detachment zone. In fact, you present data that indicates cold basal conditions in that zone.

*In reality we stated that the basal portion of the sliding surface—excluding the vertical and lateral ice scarps—consisted of three different units (see Fig. 9B): (a0) glacier ice, where failure occurred along an internal foliation; (s0) a thin, glassy, cryoconite-enriched ice layer, interpreted as regelation ice; and (e0) bedrock, where the failure occurred directly at the ice–bedrock interface. In summary, at the time of failure, the glacier was certainly frozen to the bedrock above the failure zone, to the west, and along its western front. Partly melting also induced by water forcing its way along the glacier base probably occurred at unit e0. The sliding surface of the Altels glacier failure (Failletaz et al., 2011) also consisted of different units, some of which contained residual ice still frozen to the bedrock.*

l.660: this is in direct contrast to the statement above. Also, what is the evidence that the remaining ice was refrozen meltwater? It could also just be a failure within basal ice.

*See comment above and see reply to reviewer's comment l.607 and to Fig.9.*

l.674: delete 'Anyhow'

*Correction was made;*

l.708: delete 'tout court'

*Correction was made;*

l.708-12: These two statements directly contradict each other

*This may be a matter of clarity, as we personally find no contradictions. Below is an attempt to establish the sequence of events:*

*At time $t_0$, the failed glacieret was still frozen—or mostly frozen—to its bedrock.*

*At time $t_1$, prolonged heating of the rocks on the eastern shoulder, along with the regelation of meltwater in the crevasse, caused progressive warming of the ice near the glacier base.*

*At time $t_2$, the water was able to force its way along this softened ice throughout the entire eastern sector (e0) and at the crevasse bottom (s0).*

*This should not be interpreted as the presence of a continuous water layer at the glacier base across the entire e0 sector. Instead: water that has made its way to the glacier base, likely driven by the overlying*

*ice pressure, appears to form an initial distributed drainage system. This system can be visualized as a micro-scale network of interconnected tiny rivulets—at times no more substantial than linked droplets—sparsely flowing directly over the bedrock.*

*We hope this clarifies the apparent contradictions.*

l.712: Why do you consider this most likely? As stated in the overall comments, I think there is a slightly different scenario that I personally find more likely, as it avoids some of the stated contradictions.

*See comment above. The scenario suggested by the reviewer in the overall comments is fully consistent with ours. We are simply attempting a more in-depth reconstruction through a careful analysis of the available data.*

l.717: I think you mean 'significantly' rather than 'largely'? In this context largely would mean 'mostly'

*Correction was made;*

l.721: I have a hard time imagining a scenario of hydraulic jacking without simultaneously losing basal friction (see overall comment)

*We addressed this issue in the general comments.*

l.724: Is that really surprising? It is the surface at which you apply the jacking

*We are referring to the vertical scarp.*

l.735: largely -> significantly

*Correction was made;*

l.769: is this known?

*Images taken the day before the failure by UAV showed water flowing out the median crevasse. Post-failure images of the remaining edges of the median crevasse shows a clearly water-eroded inside with smooth rounded surfaces.*

l.775-777: I'm personally more hopeful about the possibility of seismic monitoring, because some of these processes have different spectral signatures and could potentially be distinguished seems in terms of data the identification of large water filled crevasses would be indicative of a potentially hazardous situation that warrants detailed monitoring.

*I fully agree with the reviewer, but I fear that seismic monitoring can only be applied when the potential collapse of the glacial mass threatens property or people. In all other cases, due to cost issues, it is very*

*unlikely that a local network of sensors will be set up, as is routinely done for dams or other critical infrastructures.*